# Genomic and panproteomic analysis of the development of infant immune responses to antigenically-diverse pneumococci

Nicholas J. Croucher [1] ✉, Joseph J. Campo [2], Timothy Q. Le[2], Jozelyn V. Pablo[2], Christopher Hung[2], Andy A. Teng[2], Claudia Turner[3,4], François Nosten [4,5], Stephen D. Bentley [6], Xiaowu Liang[2], Paul Turner[3,4] & David Goldblatt [7]

*Streptococcus pneumoniae* (pneumococcus) is a nasopharyngeal commensal and respiratory pathogen. This study characterises the immunoglobulin G (IgG) repertoire recognising pneumococci from birth to 24 months old (mo) in a prospectively-sampled cohort of 63 children using a panproteome array. IgG levels are highest at birth, due to transplacental transmission of maternal antibodies. The subsequent emergence of responses to individual antigens exhibit distinct kinetics across the cohort. Stable differences in the strength of individuals' responses, correlating with maternal IgG concentrations, are established by 6 mo. By 12 mo, children develop unique antibody profiles that are boosted by re-exposure. However, some proteins only stimulate substantial responses in adults. Integrating genomic data on nasopharyngeal colonisation demonstrates rare pneumococcal antigens can elicit strong IgG levels post-exposure. Quantifying such responses to the diverse core loci (DCL) proteins is complicated by cross-immunity between variants. In particular, the conserved N terminus of DCL protein zinc metalloprotease B provokes the strongest early IgG responses. DCL proteins' ability to inhibit mucosal immunity likely explains continued pneumococcal carriage despite hosts' polyvalent antibody repertoire. Yet higher IgG levels are associated with reduced incidence, and severity, of pneumonia, demonstrating the importance of the heterogeneity in response strength and kinetics across antigens and individuals.

Lower respiratory tract infections, such as pneumonia, are the leading infectious cause of morbidity and mortality in children[1]. *Streptococcus pneumoniae* (the pneumococcus)[2] causes both invasive pneumococcal disease (IPD) and a substantial proportion of bacterial community-acquired pneumonia[3]. The highest incidence of these diseases occurs in the first year of life[1], and declines as the infant immune response strengthens[4]. The ability of pneumococci to cause IPD depends upon their expression of a polysaccharide capsule[5], of which there are over

[1]MRC Centre for Global Infectious Disease Analysis, Department of Infectious Disease Epidemiology, School of Public Health, Imperial College London, London W12 0BZ, UK. [2]Antigen Discovery Inc, 1 Technology Drive, Irvine, CA 92618, USA. [3]Cambodia Oxford Medical Research Unit, Angkor Hospital for Children, Siem Reap 9V54+8FQ, Cambodia. [4]Centre for Tropical Medicine and Global Health, Nuffield Department of Medicine, University of Oxford, Oxford OX3 7BN, UK. [5]Shoklo Malaria Research Unit, Mahidol–Oxford Tropical Medicine Research Unit, Faculty of Tropical Medicine, Mahidol University, Mae Sot 63110, Thailand. [6]Parasites & Microbes, Wellcome Sanger Institute, Wellcome Genome Campus, Hinxton, Cambridge CB10 1SA, UK. [7]Great Ormond Street Institute of Child Health, University College London, London WC1N 1EH, UK. ✉e-mail: n.croucher@imperial.ac.uk

100 immunologically-distinct serotypes known[6]. Children under 24 months old (mo) are unable to mount an effective T cell-independent response to this capsule[7], and therefore early adaptive immune reactions, targeting pneumococcal proteins, are likely important in infants' declining susceptibility to infection[8,9].

Anti-protein responses develop during asymptomatic nasopharyngeal carriage of pneumococci[10–14]. Despite the accumulation of mucosal antibodies targeting pneumococcal proteins, primarily of the immunoglobulin A (IgA) isotype, children repeatedly reacquire many strains of pneumococci throughout their life[14,15]. Yet colonisation duration declines with age, and experimental human carriage studies have found colonisation protects against rechallenge with the same isolate[13,16], suggesting individuals acquire protection against carriage[17,18]. However, the relative importance of anti-protein and anti-capsular antibodies, as well as CD4[+] T cell responses, is unknown[8,19–23]. Although examples of protein-targeting antibodies preventing carriage of pneumococci have been identified[24–26], multiple analyses have not found any general association between antibody levels and colonisation[24,25,27–31]. Furthermore, immunostimulatory protein-based vaccines have not significantly reduced carriage[19,32].

There is stronger evidence that systemically-circulating antibodies, primarily of the immunoglobulin G (IgG) isotype, protect against pneumococcal disease[8,13,33]. The effectiveness of anti-protein IgG has been demonstrated therapeutically by intravenous immunoglobulin (IVIG) transfers to individuals with an impaired immune response[34]. Detailed analysis of the IgG transferred in IVIG revealed healthy adult immune responses recognise a consistent set of pneumococcal proteins[33]. These antibody-binding targets (ABTs) have been characterised using a panproteome array, which assays responses to thousands of proteins identified across a diverse pneumococcal population[35]. This demonstrated IgG responses commonly recognised adhesins; proteins involved in the production of the bacterial cell wall; degradative enzymes; and solute binding proteins (SBPs), which enable bacteria to acquire nutrients[36]. Many of these proteins are highly conserved across the pneumococcal population, whereas other ABTs, such as the type 2 pilus, are only present in a subset of isolates[36].

Approximately half of the ABTs were distinct variants of four diverse core loci (DCL): pneumococcal surface protein A (PspA); pneumococcal surface protein C (PspC; also named CbpA, Hic or SpsA); zinc metalloprotease A (ZmpA), and zinc metalloprotease B (ZmpB). The DCL are ubiquitous across the population, but highly variable in their sequences[36]. Both PspC and ZmpA are IgA-binding proteins: PspC inhibits the binding of human IgA to bacterial antigens[37,38], whereas ZmpA is a protease that degrades IgA[39]. PspC also inhibits immune responses through binding proteins in the complement cascades[38], a function shared by PspA[40,41]. The function of ZmpB remains unknown[39,42], although it is also likely to modulate immune responses, as the absence of this protein resulted in reduced inflammatory responses in a mouse model of disease[43]. Panproteome array analyses have demonstrated both whole cell vaccines[16,44] and experimental human carriage[16] boost responses to DCL proteins and other antigens in healthy adults in high-income countries. However, anti-pneumococcal protein antibodies are most important in protecting against disease in infants, in whom our understanding of the early emergence of antibody responses is limited to relatively few antigens[28,29,45–48].

In this work, we apply the panproteome array to fully characterise the antibody dynamics in a prospectively-sampled cohort of children in the Maela camp for refugees in rural Thailand[14]. During the first 24 months of life, nasopharyngeal colonisation of these individuals was assessed at monthly intervals while healthy[14], and during each episode of clinical pneumonia. Pneumococci isolated from these swabs have been extensively characterised through serotyping[49], genotyping[50], single isolate genomics[51–53] and deep sequencing[54]. Serum samples were also regularly collected, enabling ELISA assays that demonstrated

overall IgG responses were highest at birth, fell to a nadir at 6 mo, then rose as with age[45], consistent with previous studies[26,28,29,47]. Such age-associated changes in IgG levels are typically reflective of similar patterns in IgA concentrations[48]. By synthesising genomic and panproteome array data, this cohort provides a high-resolution view of how individual-level exposure to pneumococci affects the emergence of multiple adaptive immune responses in children.

## Results

### Individual-specific immune profiles develop in the first year of life

The panproteome array was used to assay blood samples from the births of 63 children (both from the mother and the umbilical cord), then from the child at 6, 12, 18 and 24 mo. One sample failed quality control standards, resulting in 377 datasets (Supplementary Data 1). The array measurements of IgG binding were generally strongly correlated with previous ELISA assays of antibody titres to candidate vaccine antigens, which used a subset of the same serum samples (Supplementary Fig. 1). An unsupervised kernel-based clustering algorithm classified the datasets into six groups (Supplementary Figs. 2 and 3). Two clusters accounted for all samples collected from children ≥12 mo, hence were classified as infant clusters. The other four clusters predominantly consisted of maternal blood and umbilical cord samples, and were therefore classified as neonatal clusters (Fig. 1a).

Three neonatal clusters were distinguished by their high, intermediate and low levels of IgG binding (Supplementary Fig. 4). The fourth was termed "infant-like", as it resembled the two infant clusters (Supplementary Fig. 3), which were themselves differentiated by their high or low IgG binding levels. Across ages and ABT types, the IgG binding levels in the "infant high" cluster were 1.53-fold greater than those of the more common "infant low" cluster (Supplementary Fig. 5). This ratio was estimated as a 1.49-fold difference using the ELISA antibody titres, confirming this distinction was not an artefact of the array analysis (Supplementary Figs. 5 and 6). By contrast, there was little evidence of a substantial difference in the anti-capsular IgG levels between clusters (Supplementary Fig. 7). The proportions of individuals belonging to the infant low immune cluster did not significantly change between the ages of 6 and 24 mo (two-tailed Fisher's exact test, $N = 251$, $p = 0.67$). Rather, belonging to the low IgG infant cluster at 6 mo was significantly predictive of still belonging to this cluster at 24 mo (two-tailed Fisher's exact test, $N = 63$, odds ratio $= 44.9$, $p = 7.6 \times 10^{-9}$). Hence individuals differed in the strength of their antibody response to pneumococci early in life, and these distinctions persisted as the immune system matured.

This clustering could reflect environmental or physiological heterogeneity within the cohort. However, the categorisation did not reflect differences in birth weight (Fig. 1b and Supplementary Fig. 8), nor was there a significant association with infant sex (Fig. 1c and Supplementary Fig. 9). Similarly, the four ethnicities represented in the cohort were associated with significant differences in IgG levels at some timepoints (Supplementary Fig. 10), but the most common identity, S'gaw Karen, was not congruent with the immune clustering. Furthermore, there was no evidence that the clustering was caused by air quality. Most homes were heated by wood or charcoal fires, but the type of fuel burned did not affect infant antibody responses (Supplementary Fig. 11). Additionally, 27.0% of the mothers smoked, and many IgG responses to pneumococcal proteins were lower in the maternal blood and umbilical cord samples from these births (Supplementary Fig. S12). However, the differences disappeared as the children aged, and maternal smoking was not associated with the immune clustering. Hence available epidemiological data could not explain the differences in the emergence of the antibody response.

Concurring with the distinction between neonatal and infant clusters, t-distributed stochastic neighbour embedding (t-SNE) found no strong similarity between the IgG profiles sampled from an

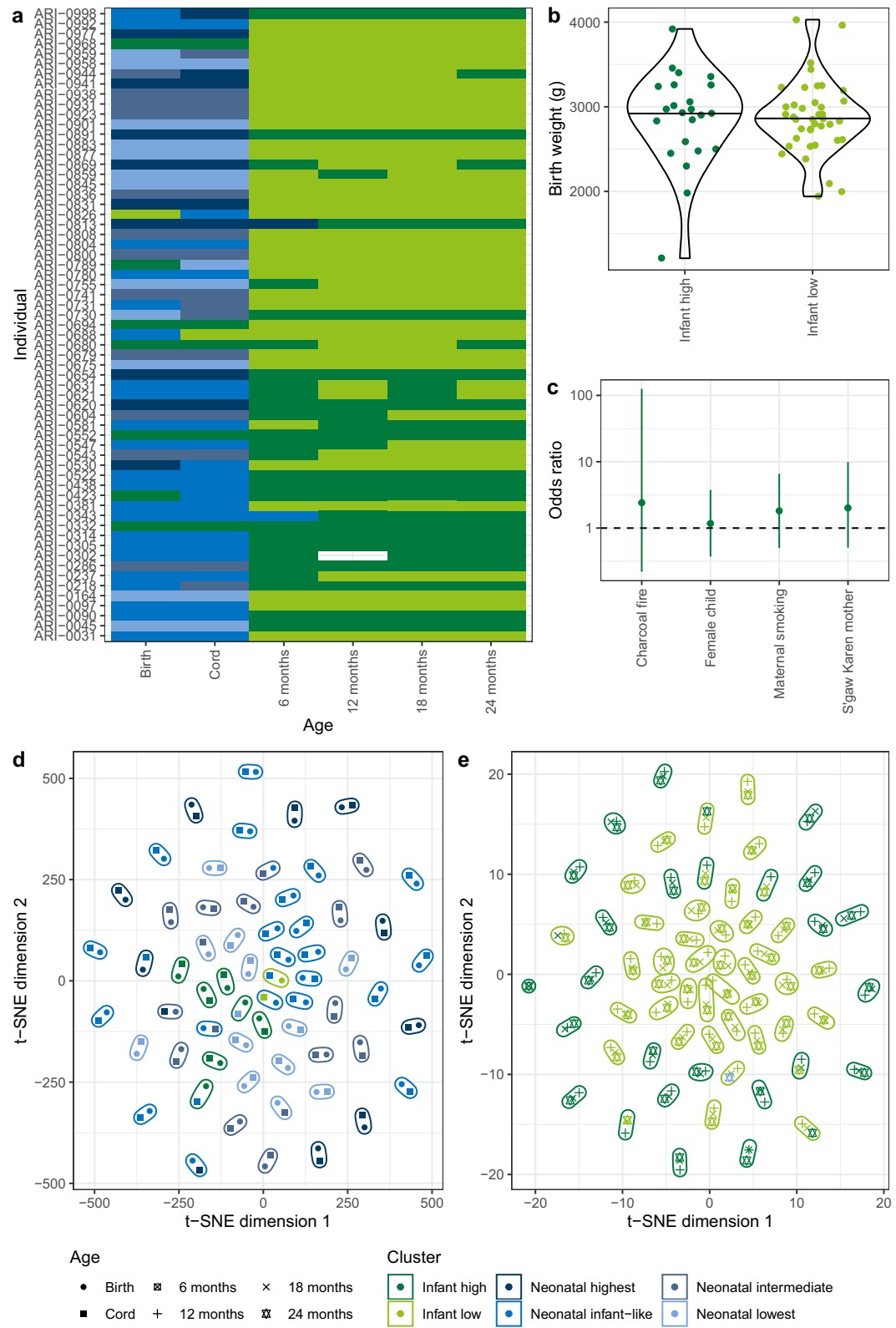

individual at birth and ≥12 mo (Supplementary Figs. 13 and 14). However, a t-SNE of only the umbilical cord and maternal blood IgG resolved the dataset into pairs of samples from each birth (Fig. 1d). Hence both neonatal samples likely represent each mother's unique set of antibodies being transplacentally transmitted into the child. Similarly, a t-SNE of sera from older children clustered together all samples from 12, 18 and 24 mo by individual (Fig. 1e), indicating each child generated a unique anti-pneumococcal response. This pattern was not an artefact of batch effects (Supplementary Fig. 15). Most sera from 6-month-old children also co-clustered with later samples from the same individual (Supplementary Fig. 16). Hence the majority of children generate polyvalent responses to pneumococci by 6 mo, and the whole cohort had produced a detectably unique anti-pneumococcal IgG profile by 12 mo.

**Fig. 1 | Variation in immune responses across individuals. a** A grid showing the assignment of panproteome array datasets ($n = 377$) to immune clusters. Each row corresponds to an individual ($n = 63$). Each column corresponds to a sample type ($n = 6$). The cells are coloured according to the cluster to which the dataset was assigned. **b** Violin plot showing the birth weights of children assigned to the infant high ($n = 23$) or infant low ($n = 40$) immune clusters at 24 mo. **c** Odds ratios for the associations between different environmental and physiological factors and the assignment of children to the infant high, rather than infant low, immune cluster at 24 mo. Each factor was converted to a binary categorisation of whether individuals ($n = 63$) belonged to the modal category or not, and a two-tailed Fisher's exact test

used to conduct each univariate analysis. The points represent the maximum likelihood estimates of the odds ratios, and the error bars show the 95% confidence intervals. **d** Scatterplot showing a t-SNE projection of the variation between samples taken from the umbilical cord, and from the mother at birth. The shape of each point shows the sample type, and the colour of the point shows the cluster to which it was assigned. Each set of samples from an individual are grouped within an ellipse. **e** Scatterplot showing a t-SNE projection of the variation between samples taken from 12 mo onwards. Data are shown as in (**d**). Source data are provided as a Source Data file.

## Antibody levels and kinetics vary across antigen functional classes

This transition from maternally-derived to endogenously-generated antibody profiles between birth and 12 mo was associated with substantial decline in IgG binding to all ABT types except ZmpB (Fig. 2a). Antibody levels to all non-ZmpB ABTs rose steadily from 12 mo onwards. The responses to ABTs of the same type were sufficiently consistent across the cohort to identify differences in their kinetics (Fig. 2a). Whereas infant IgG against SBPs quickly stabilised at levels approximately half those in the maternal blood, IgG against PspA and PspC fell until 12 mo, before rising in older infants. Furthermore, there were also examples of ABTs, particularly among the adhesins, that were bound by high levels of IgG in the birth samples, but elicited no substantive response in infants.

The antigens that failed to drive early endogenous responses were identified by comparing the IgG responses in the maternal birth and 24 mo infant samples (Fig. 2b). They included large, repetitive adhesins such as the glycosylated serine-rich repeat protein PsrP[36,55], the glycan-binding protein PfbA[56], and multiple variants of the collagen-like antigen PclA[36,57]. Yet similar deviations were also evident for ABTs that are typically intracellular (e.g. AcoL, GroEL). These antigens all shared the property of only stimulating intermediate-strength responses in adults, suggesting they were not sufficiently immunogenic to trigger IgG generation in infants.

By contrast, many highly immunogenic proteins elicited responses of similar strength across the cohort at both timepoints. This consistent emergence of high-level responses to a concordant set of ABTs in all individuals was reflected by the broad set of proteins contributing to the differentiation of immune clusters (Supplementary Fig. S17). Hence the heterogeneity across the population principally represented quantitative differences in the IgG response to a common set of epitopes, rather than recognition of distinct sets of antigens.

To analyse these differences, thirteen different linear mixed-effects models were fitted to the IgG data, each including the individual as a random variable (Supplementary Table 1). The model best representing the data included fixed effects that were consistent with antigen kinetics varying both by antigen type and the individual's cluster (Fig. 2c and Supplementary Figs. 18 and 19). Equivalent models that substituted the impact of immune cluster on IgG levels for environmental and physiological variables were less effective at explaining the differences in antibody responses, confirming the lack of association between these factors and the detected heterogeneity across individuals (Supplementary Fig. 20 and Supplementary Table 2). Despite the differing strength of the overall IgG response between clusters, the rank order of responses to ABT types was concordant across them (Supplementary Fig. 21). The exception was the levels of antibodies recognising ZmpA in the infant clusters, with IgG binding particularly reduced in the infant low cluster at 6 mo, followed by some subsequent narrowing of the difference with age (Supplementary Fig. S22). Hence the differences in the kinetics with which IgG accumulated to functionally-distinct pneumococcal antigens were consistent across the cohort, despite quantitative variation in individuals' overall levels of antibody responses.

## Strong early immune responses target the conserved N terminus of ZmpB

The IgG responses to ZmpB were atypical in changing little with age, due to consistently high binding of all variants' N terminal regions (Fig. 3a and Supplementary Fig. 23). ZmpB N terminal probes provoked the strongest IgG interaction in 40% of serum samples (Supplementary Fig. 24), including the majority of datasets at 12 mo (Fig. 3b). By contrast, IgG binding to the other ZmpB probes showed the trends expected of a maturing immune response, as did probes corresponding to the paralogous ZmpA, ZmpC and ZmpD proteins[39] (Fig. 3a and Supplementary Fig. 25). As ZmpA engages the invariant region of its IgA substrate through high affinity interactions that do not involve the C terminal region[58,59], and the ZmpB N terminus shares structural domains with the immunomodulatory PspC protein (Supplementary Fig. 26), it seemed likely that ZmpB may directly bind antibodies outside of their variable antigen recognition domains. To test this, the binding of ZmpA and ZmpB probes to four human monoclonal IgG that were specific to capsular polysaccharides was assayed. This found no detectable binding between ZmpB and the antibodies (Fig. 3c). Therefore the strong IgG binding of ZmpB represents a rapidly-emerging adaptive immune response to the protein.

As 6 mo infants had developed a broad response to all ZmpB variants, despite not having been exposed to many alleles, the IgG must recognise an epitope shared by all the representatives on the array. Despite the overall diversity of the ~2000 amino acid (aa) protein, the N terminal 279 aa are highly conserved (Supplementary Fig. 27). Previous antibody fingerprinting analysis of a single variant found this was the region bound by immunoglobulins[60]. Hence this highly immunostimulatory polypeptide is broadly recognised across pneumococci by the strongest emerging adaptive immune response of young children.

## Maternal and infant antibody levels are correlated

The stable assignment of individuals to immune clusters over the duration of the study (Fig. 1a) suggests a subset of infants accumulated anti-pneumococcal antibodies at an accelerated rate during their first 6 months. Retrospective analysis of each timepoint demonstrated that individuals assigned to the infant high cluster at 24 mo had significantly greater IgG levels than those from the infant low cluster even in the maternal birth and umbilical cord samples (Supplementary Fig. 28). To quantify the importance of maternal IgG levels to infant immunity, the correlation of antibody binding levels to each protein across individuals was calculated between the maternal birth sample, and that from each other timepoint (Fig. 4a). Most ABTs exhibited Pearson $R$ values above 0.75 in the comparison of the paired cord and maternal birth sera, consistent with the high similarity between these samples (Fig. 1d). These $R$ values fell to approximately 0.25 by 6 mo, and plateaued in the range 0.1–0.25 from 12 mo onwards. Hence the association between maternal and infant IgG responses is weak at the resolution of an individual antigen (Supplementary Fig. 29), but combining associations across many ABTs demonstrates higher maternal antibody levels are associated with elevated infant IgG responses in the first years of life.

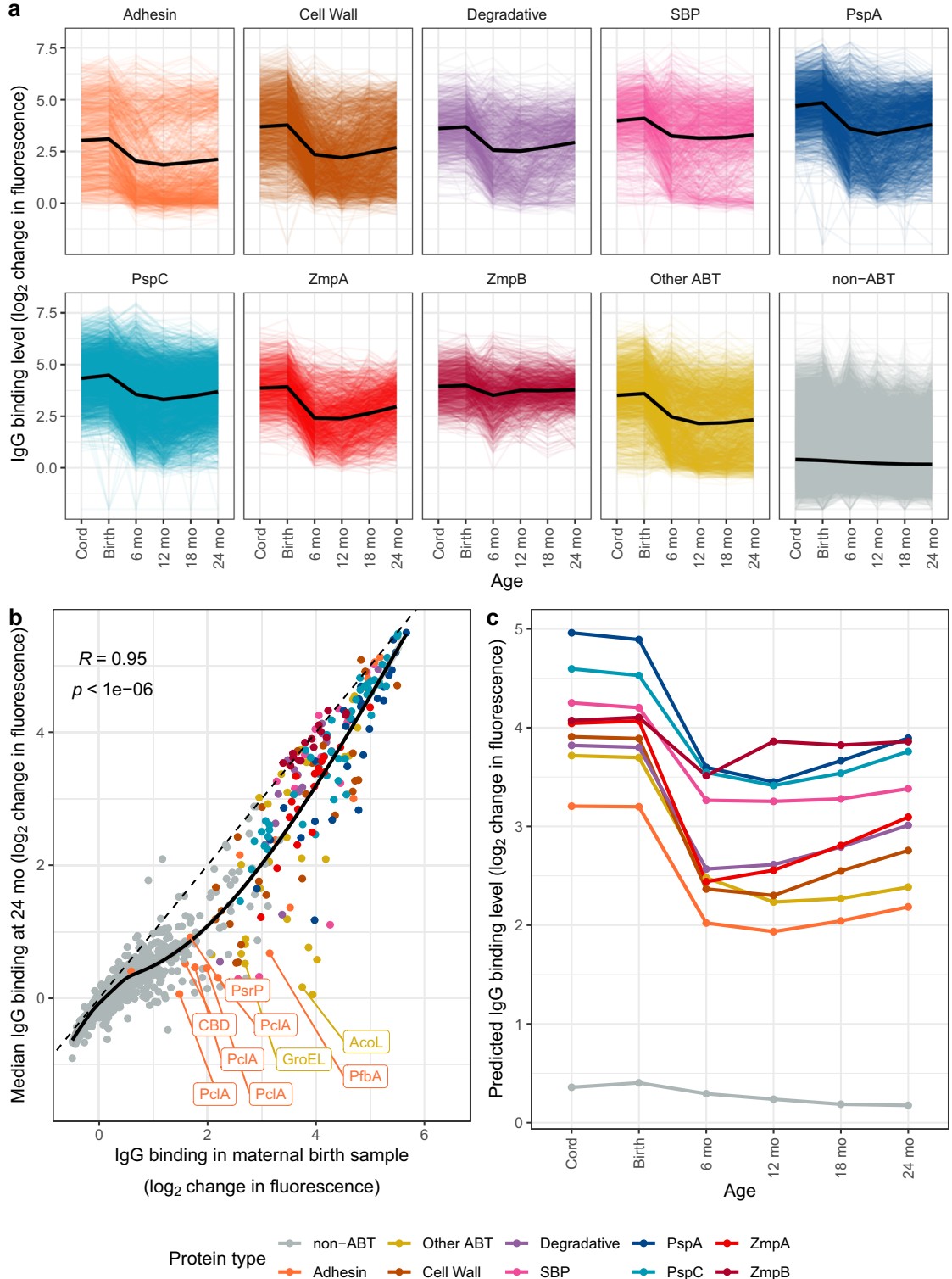

**Fig. 2 | Changing IgG responses with age.** Each protein was classified into a functional type: adhesin ($n = 16$); cell wall metabolism ($n = 26$); degradative enzymes ($n = 15$); solute binding proteins (SBP, $n = 16$); pneumococcal surface protein A (PspA, $n = 29$); pneumococcal surface protein C (PspC, $n = 48$); zinc metalloprotease A (ZmpA, $n = 17$); zinc metalloprotease B (ZmpB, $n = 15$); other antibody-binding targets (Other ABT, $n = 28$), and non-ABT ($n = 1563$). **a** Line plot showing the IgG binding to proteins at different ages. Proteins are grouped and coloured by their type. Each line shows the response to a protein in an individual. The solid black lines show the trend with age for each type predicted by a linear mixed-effects model fitted to the 377 panproteome array datasets (Supplementary

Table 1). **b** Scatterplot showing the relationship between the median IgG binding to each protein ($n = 1773$) in the maternal birth samples and the samples from 24 mo children. Points are coloured by their functional type. The dashed line shows the line of identity, and the solid line shows the relationship between the observations calculated by Loess smoothing. The abbreviation "CBD" refers to a polypeptide consisting of repeated choline-binding domains. The significance of the correlation between the two sets of measurements was assessed using a two-tailed Pearson's correlation test. **c** Line plot showing the changes in IgG binding to protein types over time inferred by the linear mixed-effects model. Source data are provided as a Source Data file.

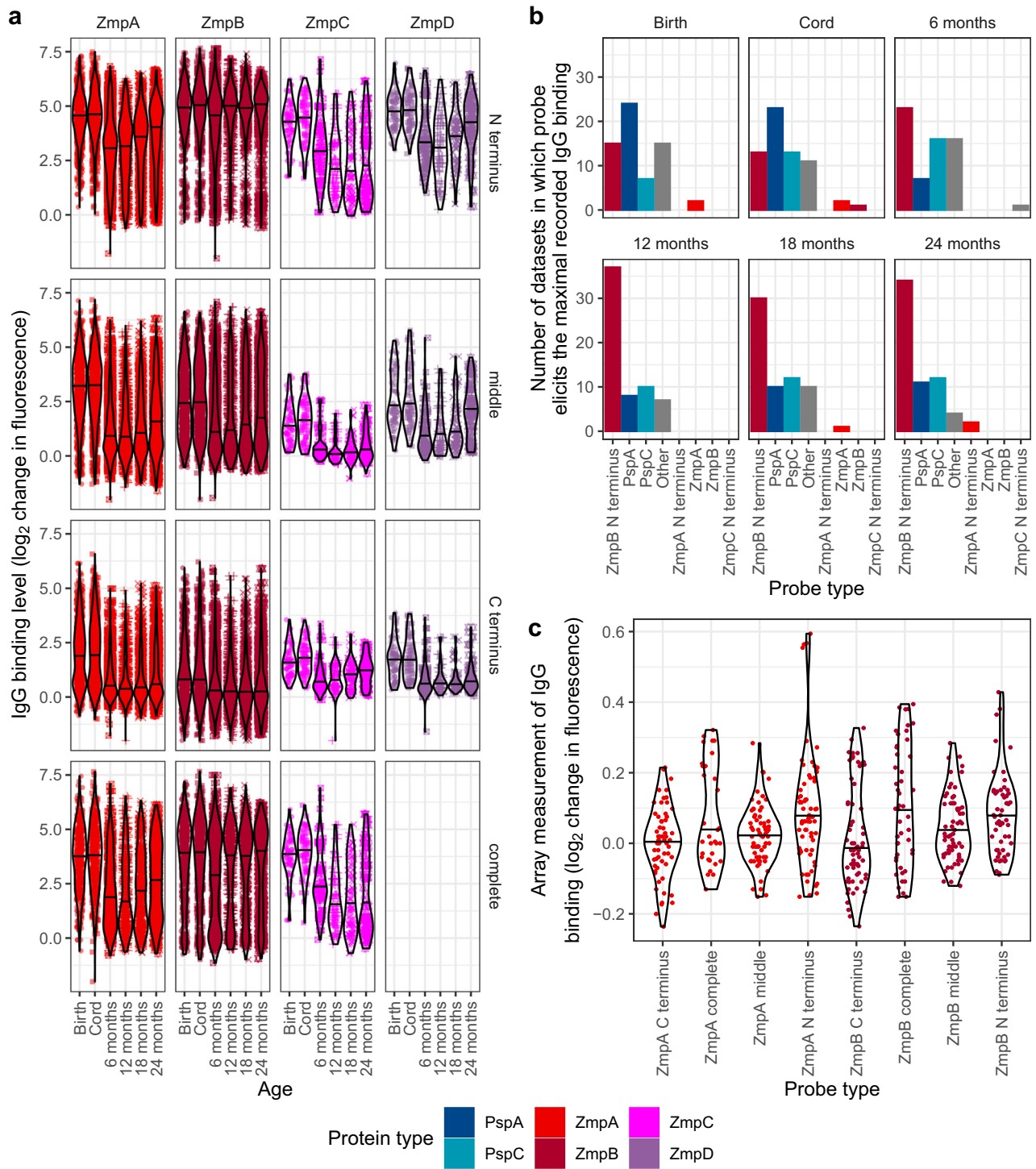

**Fig. 3 | The rapidly-emerging adaptive immune response to the N terminus of ZmpB. a** Violin plot showing IgG binding to the array probes representing the zinc metalloproteases ZmpA ($n = 56$), ZmpB ($n = 56$), ZmpC ($n = 4$) and ZmpD ($n = 3$). The horizontal line across the violin shows the median value of the raw data, which are represented by the individual points. Each plot shows the binding to a type of probe across all variants of the corresponding protein. No complete version of ZmpD was included on the array. **b** Barplot showing the type of probe stimulating the strongest IgG binding within each serum sample, categorised by the age at which the sample was taken. **c** Violin plot showing the binding of four capsule polysaccharide-specific human monoclonal IgG to the ZmpA ($n = 57$) and ZmpB ($n = 58$) probes on the array. The absence of any immunoreactivity in this analysis demonstrates that the high level of IgG binding to the ZmpB N terminus in the serum samples is the consequence of an adaptive immune response to a specific epitope. Source data are provided as a Source Data file.

As overall IgG levels at 6 mo were predictive of subsequent clustering, the correlations between this timepoint and later samples were analysed at the antigen-specific level. These correlations were generally stronger than those using the maternal sample as the comparator (Fig. 4b). The $R$ values generally declined from around 0.50 at 6 mo to 0.25 at 24 mo, with the exception of ZmpA, which declined from around 0.65 to 0.50 over the same period. By contrast, a few proteins exhibited $R$ values close to, or below, zero. Some of these are variable in sequence, or rare, across the pneumococcal population[36]. Hence changes in IgG responses to such antigens are

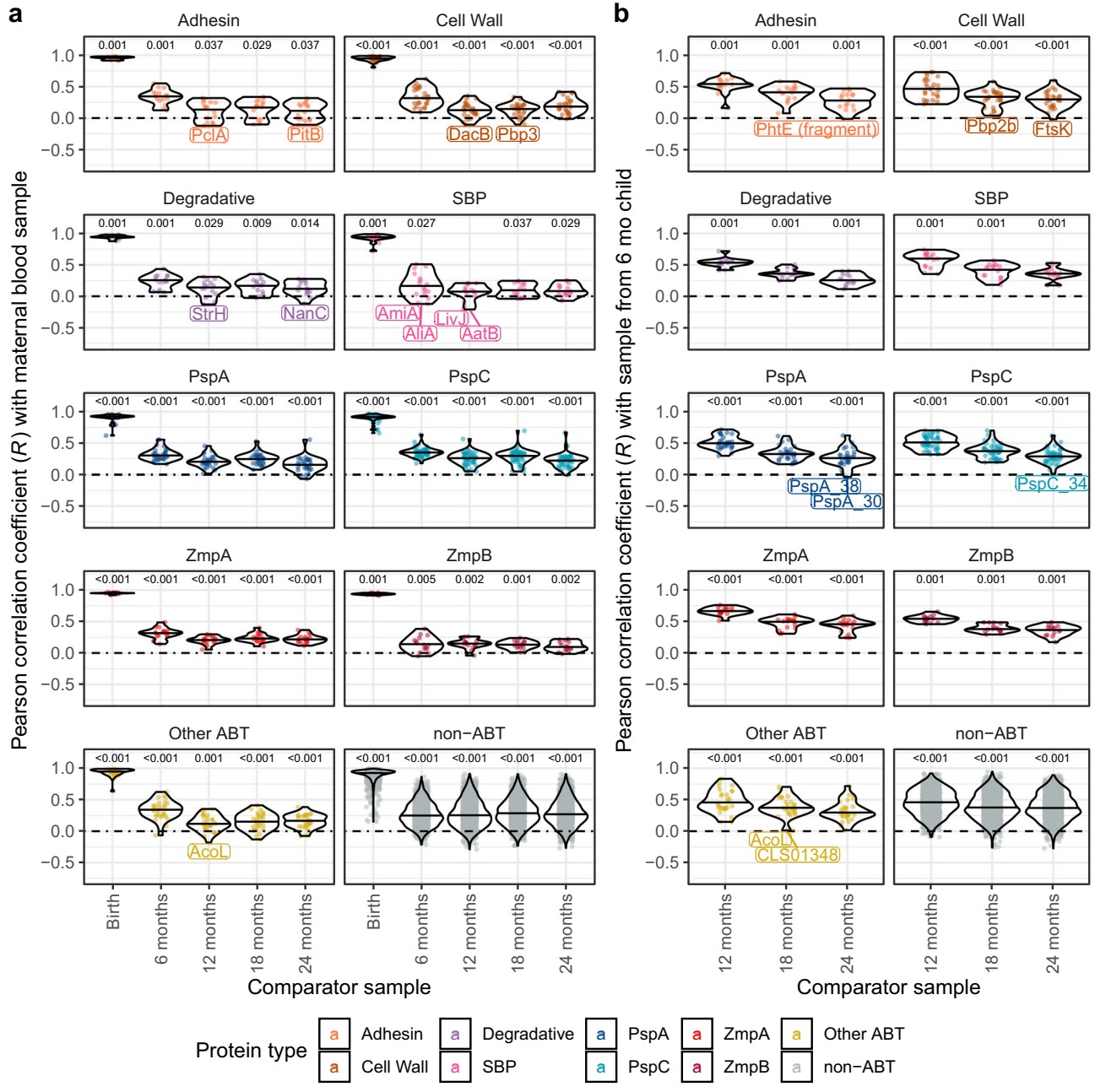

**Fig. 4 | Correlations between IgG binding levels at early and subsequent timepoints.** Violin plots showing the correlation between IgG levels at different timepoints across individuals for each protein. Each point represents the Pearson $R$ correlation coefficient for a protein across individuals at two sampled ages. ABTs for which the $R$ coefficient was negative are labelled. Proteins are grouped and coloured by their type, as displayed in Fig. 2. The data for each type is summarised by a violin plot with a horizontal line showing the median value. For each comparison between sampling ages, for each protein type, a two-tailed Wilcoxon rank sum test was used to assess whether the $R$ values deviated from the null hypothesis of no correlation in IgG responses between sampling ages, with a Holm-Bonferroni correction for multiple testing applied within each panel. Significant deviations from this null hypothesis are annotated with $p$ values above the corresponding plot. **a** Correlations between the IgG level in the maternal blood sample and the later samples. **b** Correlations between the IgG level in the sample taken at 6 mo and later samples. Source data are provided as a Source Data file.

likely to reflect individuals' history of exposure to pneumococcal proteins.

## Antibody responses broaden with antigen exposure and strengthen with age

A total of 2718 nasopharyngeal swabs had been collected from the 63 mother-child pairs in this cohort (Supplementary Fig. 30): 2534 routine surveillance swabs (84.6% of which were positive for pneumococci), and 184 associated with the diagnosis of pneumonia (90.8% of which

were positive for pneumococci; Supplementary Table 3). By combining single isolate and deep sequencing data[51–54,61] (Supplementary Fig. 31), contigs could be assembled representing DNA extracted from 1093 samples (47.2% of positive swabs), with equivalent data available for contemporaneous serotype- and genotype-matched pneumococci from Maela for an additional 1103 swabs (47.7% of positive swabs; Supplementary Tables 4–5 and Supplementary Data 2, see "Methods"). These isolates represented 89 different Global Pneumococcal Sequence Clusters (GPSCs)[62] and 56 different serotypes

(Supplementary Fig. 32). The distributions of the array proteins were inferred using a combination of graph-, mapping- and alignment-based methods (Supplementary Figs. 33 and 34; see "Methods"). The accuracy of these inferences was checked by ensuring the corresponding gene frequencies correlated with those in the population in which the proteins were originally identified[61] (Supplementary Fig. S35); that gene frequencies were consistent across single isolate and deep sequencing data[54] (Supplementary Figs. 36 and 37), and that the distribution of genes reflected the underlying population structure[36,63] (Supplementary Figs. 38–42). This analysis enabled the inference of the times at which individuals had been first exposed, and were subsequently re-exposed, to the antigens on the array.

The IgG responses to multiple ABT types were significantly greater in individuals inferred to have been exposed to an antigen at any point since birth, relative to the equivalent responses in individuals for whom such evidence was lacking (Fig. 5). An exception was the cell wall metabolism enzymes, as the only representative of this category that is not highly conserved across pneumococci was the phage-encoded lytic amidase (Supplementary Fig. 43). However, the high similarity of such proteins to the conserved cellular amidase[64] means individuals not exposed to pneumococcal prophage are still likely to have cross-immunity against the viral orthologue.

Given the importance of the first months of life in determining the stable differences observed after 12 mo, it seemed possible early exposures may shape the adaptive immune response particularly strongly, akin to the "original antigenic sin" observed with influenza[65]. However, there was no evidence that members of the infant high cluster were first colonised at an earlier timepoint, or colonised more frequently when young (Supplementary Fig. 44). Similarly, there was little evidence of stronger responses to ABTs if they were encountered in the first month (Supplementary Fig. 45), or first 6 months (Supplementary Fig. 46), of life. There was also no indication that IgG levels were transiently elevated by exposure in the month before the sample being taken (Supplementary Fig. 47), or in the 6-month interval since the previous sample (Supplementary Fig. 48). Hence responses to ABTs were stable and not strongly influenced by the timing of exposure.

To quantify the impact of exposure on the adaptive immune response, the linear mixed-effects models were modified to analyse the variation in IgG responses to ABTs from 6 mo onwards as a function of maternal antibody levels and children's history of exposure to proteins (Supplementary Table 5). For antigens to which the cohort was universally exposed by 12 mo, the best-fitting model included a term accounting for the correlation with IgG levels in the maternal birth sample (Supplementary Fig. 49). For antigens to which there was differential exposure across the cohort at 12 mo, the best-fitting model added a further term specifying whether an individual had carried a pneumococcus expressing the ABT at any point since birth, rather than requiring the exposure to occur early in life or shortly before sampling. Combining these two best-fitting models to the complementary antigen sets demonstrated they jointly reproduced the observed IgG values with similar accuracy (Supplementary Fig. 50). Both multivariate analyses produced consistent estimates of the positive contribution of maternal immunity to infant responses, and strongly correlated estimates of the strength of individuals' overall immune responses (Supplementary Fig. 51). Hence individuals maintain stable IgG repertoires recognising their earliest colonisation episodes, which expand intermittently throughout life as they encounter novel proteins.

## Seroconversion following exposure to rare antigens is rapid and stable

Eight ABTs triggered a doubling, or more, of the relevant antibody response after exposure (Fig. 6a). These included four non-DCL rare antigens: the type 2 pilus protein PitB[66] (CLS02871; present in 18.5% of

pneumococci); the neuraminidase NanC (CLS01160; present in 16.8% of pneumococci)[36]; ZmpC[67] (CLS01991; present in 5.77% of pneumococci) and ZmpD[39] (CLS02608; present in 10.0% of pneumococci). Analysing the corresponding responses across individuals in relation to the number of nasopharyngeal swabs positive for these antigens suggested IgG levels converged on a consistent maximum after two to four positive samples (Fig. 6b). Similar patterns were observed for alleles of the type 1 pilus protein RrgB[36,50,68], not classified as an ABT[36] (Supplementary Fig. 52). This suggests antibody responses quickly reach a plateau, determined by the immunogenicity of the corresponding antigen.

This is commensurate with the median IgG responses to other ABTs remaining stable after multiple exposures to pneumococci (Fig. 6b). Yet the model fits suggested mean IgG responses increased across ABT types from 12 mo onwards (Supplementary Fig. 53). This rise could represent exposure-independent effects of immune system maturation, or boosting of responses by re-exposure. To distinguish between these hypotheses, the change in IgG binding to a specific ABT in an individual between successive serum samples was compared with the difference between the individual's response, and the population-wide mean response, to the same ABT at the earlier sampling time (Fig. 7). The negative correlation between these quantities across ABT types and ages demonstrates the largest rises in IgG levels against an ABT occur in individuals previously exhibiting a low response to that antigen, relative to the population-wide mean. This is consistent with repeated encounters with pneumococci stimulating increases in any poor responses to immunogenic proteins, but not with a broad strengthening of all antibody responses with age. This is compatible with the stability in the median IgG levels to each ABT (Fig. 6b). Hence the gradual increase in population-wide adaptive immunity with age is driven by boosting of individuals' weakest responses to antigens following re-exposure.

The other four proteins to which IgG responses doubled after exposure were variants of PspA and PspC (Fig. 6b). Only PspA allele 17 was common enough to infer that IgG levels plateaued after a small number of exposures, as for PitB and ZmpD. Yet across all four DCL proteins, the median change in IgG binding following exposure was approximately zero (Fig. 6a), with rises being more strongly associated with age than exposure history (Supplementary Fig. 53). This suggested responses to each allele might not be independent, as a consequence of cross-immunity between the variants of each of the DCL.

## Cross-immunity to diverse antigens enables rapidly broadening responses

To understand this cross-immunity, the IgG responses to pairs of DCL variants were compared across all individuals at each timepoint. The IgG levels to PspA (Fig. 8) were generally highly correlated across all variants in the neonatal samples. This suggests all mothers had a broad repertoire of antibodies against the diverse representatives of this protein, with variation reflecting the strength of their overall anti-pneumococcal response (Fig. 1a). At 6 mo, patterns of correlated responses, indicating cross-immunity between variants, began to emerge from the background of waning broad maternal immunity. These blocks of correlation across subsets of alleles were clearer by 12 mo, likely reflecting strengthening endogenous adaptive responses recognising epitopes shared by only a subset of PspA proteins following exposure to one, or a few, variants in early colonisation episodes. These correlations persisted, but the cross-immunity patterns become less distinct, as children's responses broadened following exposure to additional variants. Hence the imperfect cross-immunity between alleles complicated the inference of the relationship between exposure and adaptive immune response.

The PspA proteins for which there was a detected relationship between exposure and response (Fig. 6) seemed likely to represent examples relatively unaffected by cross-immunity elicited by other

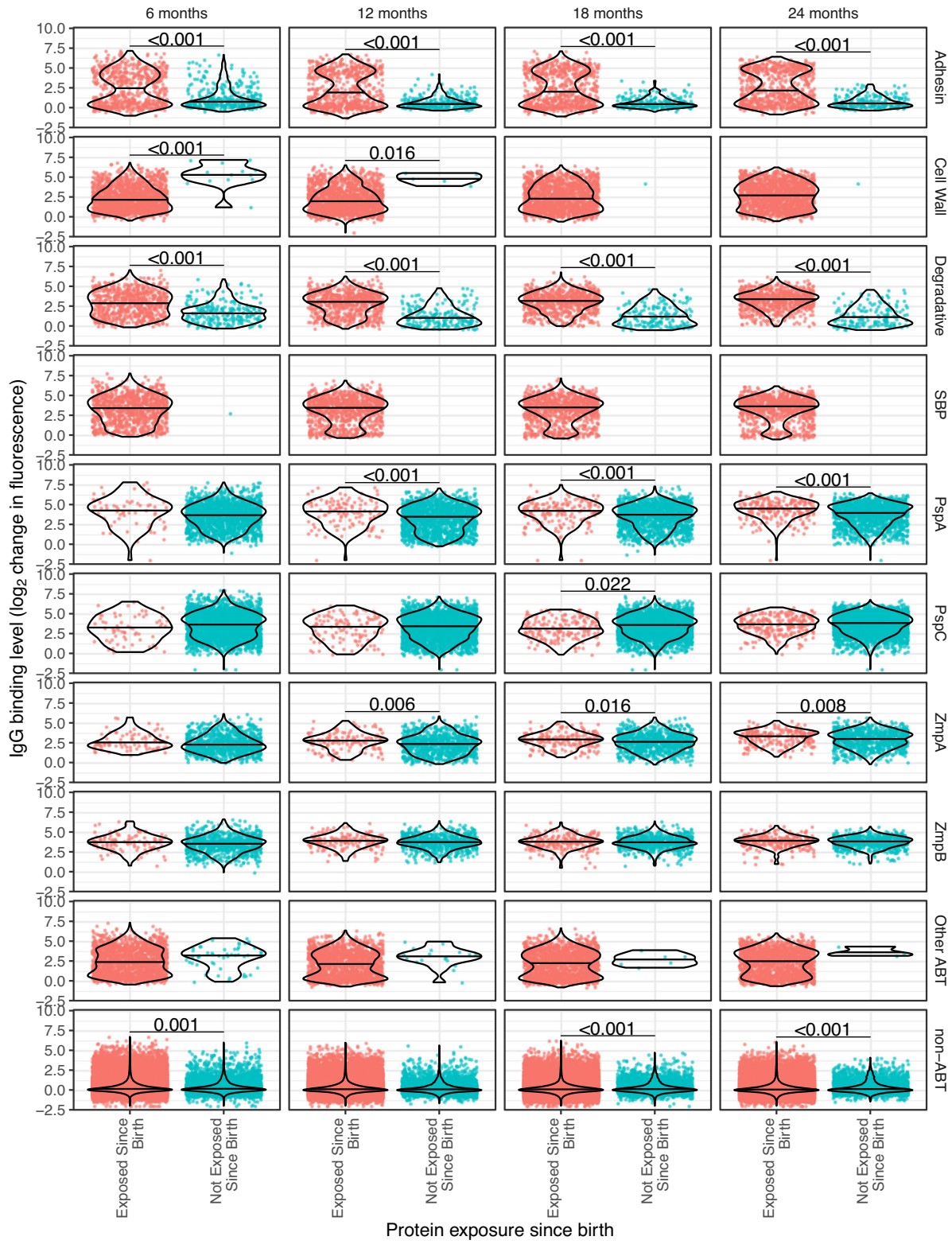

**Fig. 5 | Changes in IgG binding following exposure to proteins in carriage.** Violin plots showing the differences in IgG binding to proteins to which individuals were inferred to have been exposed since their birth, relative to those proteins for which there was no evidence of past exposure. Each row of plots corresponds to a different functional type of antigen, classified as described in Fig. 2. Each column of plots corresponds to a different timepoint. Each point corresponds to a measurement of IgG binding to a protein ($n = 1773$) in an individual ($n = 63$) at a particular timepoint. The horizontal line within each violin shows the median. The significance of differences between the measurements in each plot was calculated using a Wilcoxon rank sum test. Significant differences are denoted by horizontal bars annotated with the corresponding $p$ values. Source data are provided as a Source Data file.

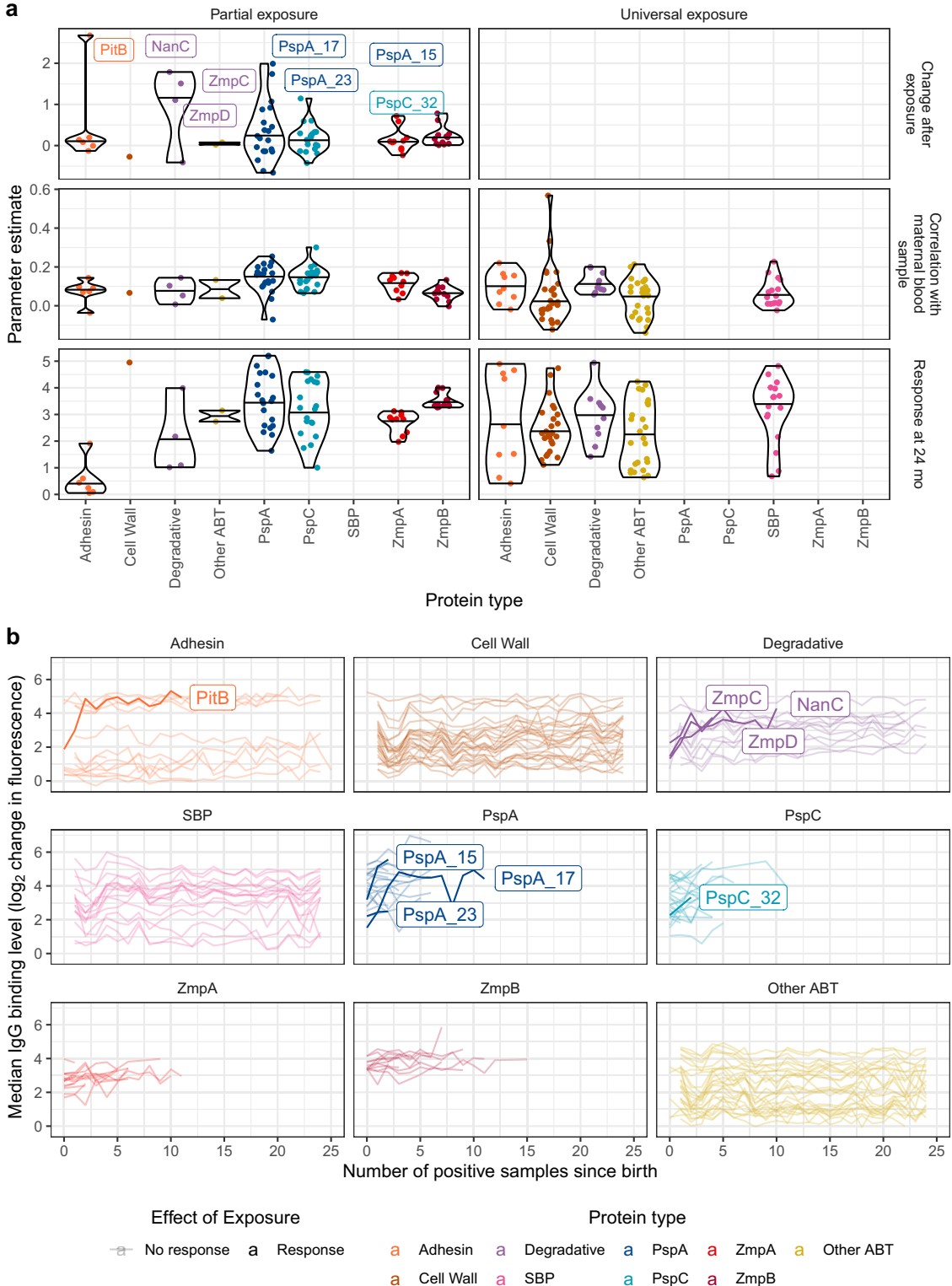

**Fig. 6 | Modelling IgG levels following exposure to antigenic proteins.**
**a** Parameter estimates from linear mixed-effects models including data on IgG levels in the maternal blood sample and individuals' history of exposure to proteins. The left column shows estimates from modelling the IgG response to proteins to which only a subset of individuals were exposed in the first year of life ($n = 75$). The right column shows estimates from a model of IgG binding to proteins to which all individuals were exposed in the first year of life ($n = 85$). The top row shows the increase in IgG binding following an individual being exposed to an antigen, which could only be estimated in the partial exposure model. The middle row shows the strength of the relationship between the IgG levels in the maternal birth sample and those at later timepoints. The bottom row shows the expected IgG response to ABTs at 24 mo, following adjustment for other factors included in the model. **b** Line plots showing the relationship between the number of pneumococcal samples isolated from an individual that encoded a protein, and the median IgG binding to the protein across individuals with the corresponding level of exposure at the time serum was sampled. The pale lines show the median IgG binding level to all ABTs ($n = 210$). The darker lines highlight the data for proteins to which IgG responses were inferred to at least double after one or more exposures in (**a**) ($n = 8$). Source data are provided as a Source Data file.

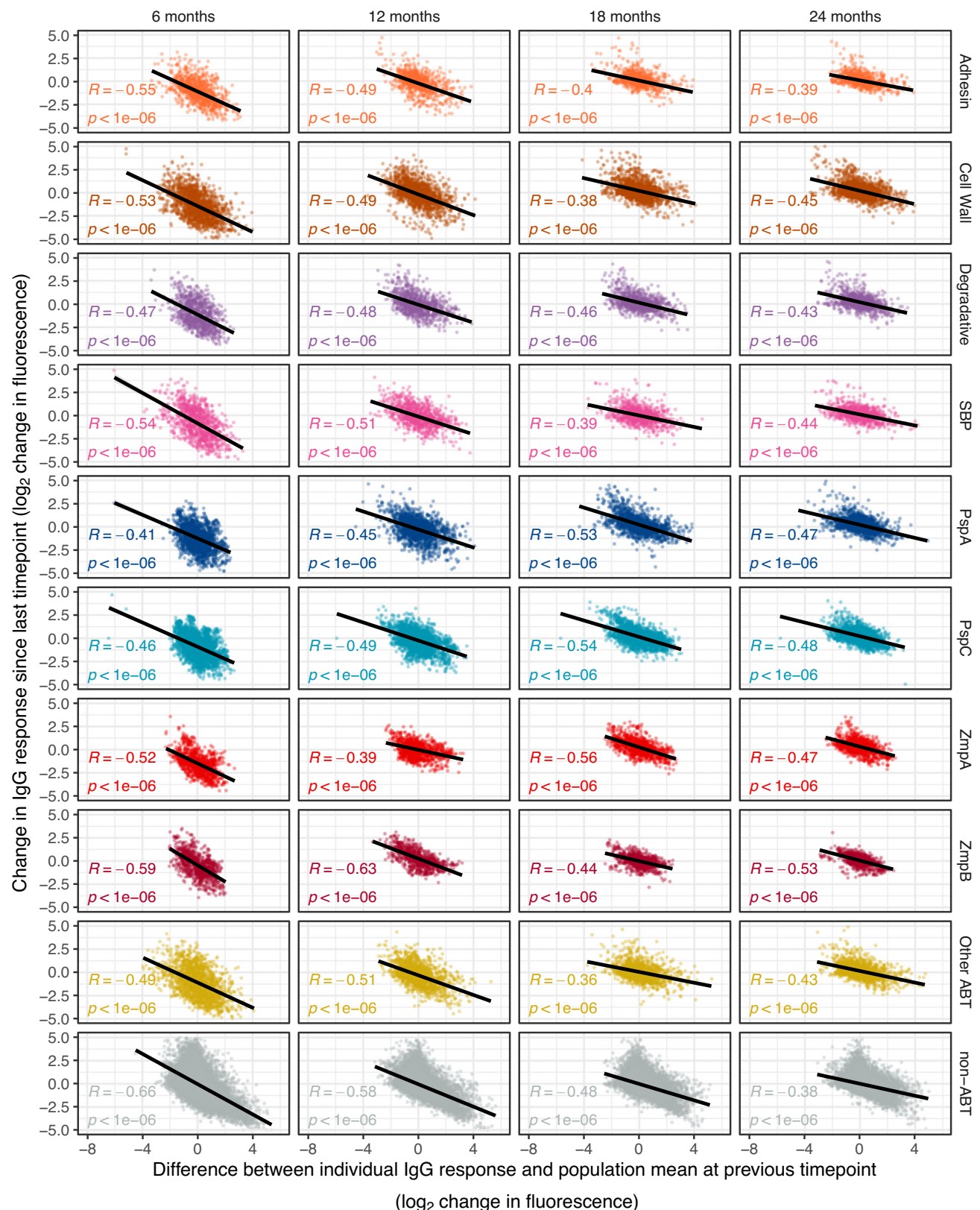

variants. This was not clearly the case for PspA 15, but responses to PspA 23 correlated only weakly with those to other PspA variants, even in the umbilical cord and maternal birth samples (Fig. 8). This may be a consequence of PspA 23's divergent sequence relative to other PspA variants[36]. Similarly, immunity to PspA 17 in older infants only correlated strongly with the highly-similar PspA 31[36]. Hence the

accumulation of antibody binding to such variants (Fig. 6b), which are minimally affected by cross-immunity, should provide a reliable measure of the immune response to individual representatives of this class of diverse proteins.

The pattern inferred for PspC was similar to that for PspA, but with the correlations implying a greater level of cross-immunity between

**Fig. 7 | Rise in IgG binding with age.** For each post-birth timepoint, the change in IgG binding to each protein (classified as described in Fig. 2) since the previous sample in each individual ($n = 63$) was calculated. This was compared with the relative strength of the IgG response in the previous sample, which was calculated by subtracting the population-wide mean response to the protein from the individual's response to the protein. Each point represents an analysis of a protein in an individual at a defined timepoint. The black lines show the linear models that were fitted to each protein type at each post-birth timepoint. The correlation statistics shown on each panel were calculated using two-tailed Pearson's correlation tests. In the context of the neonatal decline in IgG levels, the negative correlations show that the largest declines in IgG binding between birth and 6 mo reflect the proteins that were most strongly bound by IgG in the mother. In the context of the emerging endogenous antibody repertoire, the largest increases in IgG binding post-6 mo reflect relatively weak IgG responses in an individual rising towards the level observed across the rest of the cohort. Source data are provided as a Source Data file.

alleles (Supplementary Fig. 54). The variant to which a post-exposure response was inferred, PspC 32 (Fig. 6), did not provoke a notably specific response, suggesting the detected relationship may reflect a more general rise in immunity to PspC variants. Similarly, there was a high correlation across most alleles of ZmpA, with the exception of divergent patterns of IgG binding to ZmpA variants 6 and 16 (Supplementary Fig. 55), both of which were rare in Maela pneumococci (Supplementary Fig. 41). The least heterogeneity was evident in the IgG binding to ZmpB (Supplementary Fig. 56), as these measurements were dominated by the early, strong response to the conserved N terminus. Hence individuals' adaptive immune response to DCL antigens expands with age as they accumulate antibodies that can recognise multiple variants of each protein.

### More severe pneumonia is associated with lower IgG levels

The accumulation of IgG responses enabled testing of whether they protected against all-cause clinical pneumonia. Across the cohort, nasopharyngeal samples were collected during 158 episodes of non-severe pneumonia (93% positive for pneumococci), 16 episodes of severe pneumonia (69% positive for pneumococci), and 10 episodes of very severe pneumonia (90% positive for pneumococci)[69]. IgG levels were compared between children diagnosed with at least one episode of pneumonia in either the 6 months prior to sampling, or the 6 months after sampling, and the rest of the cohort (Supplementary Table 6 and Supplementary Data 3). At 6 mo, the children suffering disease in the preceding, or following, 6 months were found to have significantly lower concentrations of IgG recognising multiple ABT types relative to unaffected children (Supplementary Figs. 57 and 58). This pattern was robust to only considering pneumonia episodes classified as severe (Supplementary Figs. 59 and 60) and very severe (Supplementary Figs. 61 and 62). The divergence in responses was most consistently observed for the non-ZmpB DCL. However, these relationships were not observed in older children. Furthermore, analysing previously-published data[45] on anti-capsular immunity from this cohort found no evidence such responses were protective against pneumonia (Supplementary Fig. 63). Similarly, the frequency of colonisation was unrelated to IgG levels from 12 mo onwards, but was associated with lower IgG in the umbilical cord, and higher IgG at 6 mo (Supplementary Figs. 64 and 65). This suggested the effect of IgG on carriage and disease changed with age.

Therefore the estimation of any protective effect of IgG required adding clinical pneumonia episodes to the linear mixed effects models as an age-dependent fixed effect. This extension improved the fit of linear mixed effects models (Supplementary Fig. 66 and Supplementary Table 6), while replicating the previous parameter estimates of the simpler models (Supplementary Figs. 67 and 68). Applying this structure to all pneumonia severities, and colonisation frequency, quantified the varying relationship between adaptive immunity and diagnoses prior, and subsequent, to serum sampling across ages (Fig. 9a and Supplementary Table 7). At 6 mo, lower anti-pneumococcal IgG levels were associated with increasing severity of pneumonia in the prior, and following, 6 months. However, umbilical cord IgG levels were not associated with protection in early life. This likely reflects both the clustering of cases close to 6 mo (Fig. 9b), at the nadir of IgG levels, and the diverse aetiology of pneumonia. By contrast, frequent colonisation in the first 6 months of life was associated

with lower IgG levels in the umbilical cord sample, but higher IgG levels at 6 mo. Colonisation frequency ceased to be associated with differential IgG levels as the endogenous response strengthened with age (Fig. 9c). Analogously, very severe pneumonia was associated with low IgG concentrations in the preceding serum sample at 6 and 18 mo, but these antibody levels rose to become normal post-disease. This suggests low antibody levels, particularly common at 6 mo, increase susceptibility to pneumococcal colonisation and pneumonia, both of which are sufficiently immunostimulatory to elevate responses towards the population-wide average.

## Discussion

These data provide a detailed view of the emergence of adaptive immune responses to the pneumococcus over the first 2 years of life in a population suffering a high burden of pneumonia[14]. The analysis identified three processes, with distinct kinetics, that contributed to the emergence of the overall anti-pneumococcal protein IgG repertoire. The first process was the response to common, non-repetitive surface-exposed proteins. These were already recognised by strong antibody responses at 6 mo, following an individual's first carriage episodes. Each ABT was associated with a characteristic cohort-wide level of IgG response. This suggests antigens have an intrinsic level of immunogenicity. Further increases in the IgG levels with age were driven by repeated exposures to these common proteins boosting any responses that were weak relative to the immunostimulatory nature of the ABT. Hence healthy children should consistently generate a polyvalent response to a predictable set of pneumococcal proteins by 12 mo.

The second process was the expansion of antibody repertoires as rarer surface-exposed, non-repetitive proteins were encountered. Even in children under 24 mo, the high frequency of colonisation meant such seroconversion events were only detectable for proteins present in <20% of pneumococci, such as DCL variants. Generally, IgG binding did not increase further after 2–4 positive samples, suggesting just one or two carriage episodes of a few weeks each[70] would induce maximal antibody responses to many ABTs. This is consistent with both the rapid emergence of responses to common proteins, and responses depending more on protein immunogenicity that the timing of exposure. The distribution of one rare protein to which seroconversion was observed, ZmpD, was found to be negatively correlated with age in a previous carriage study[35]. This is consistent with pneumococci expressing this protein preferentially colonising younger children, who have yet to acquire immunity to this antigen.

The same analysis also identified PclA as being enriched in pneumococci colonising younger children[35], and two other cohorts found another large repetitive protein, PsrP, was also associated with carriage at earlier ages[71]. Immunity to such large repetitive surface ABTs developed through a third process, which did not generate substantial IgG responses in infants under 24 mo. The slow-emerging immune responses to these antigens may be primarily T cell-independent[3,7,72], as their structures may make them difficult for antigen presenting cells to process[36,73], but they likely present repetitive, polyvalent epitopes to B cells[74], akin to pneumococcal polysaccharides. Yet such a mechanism is unlikely to fully explain the structurally diverse set of ABTs to which responses at 24 mo were disproportionately weak compared with those in the mother. These

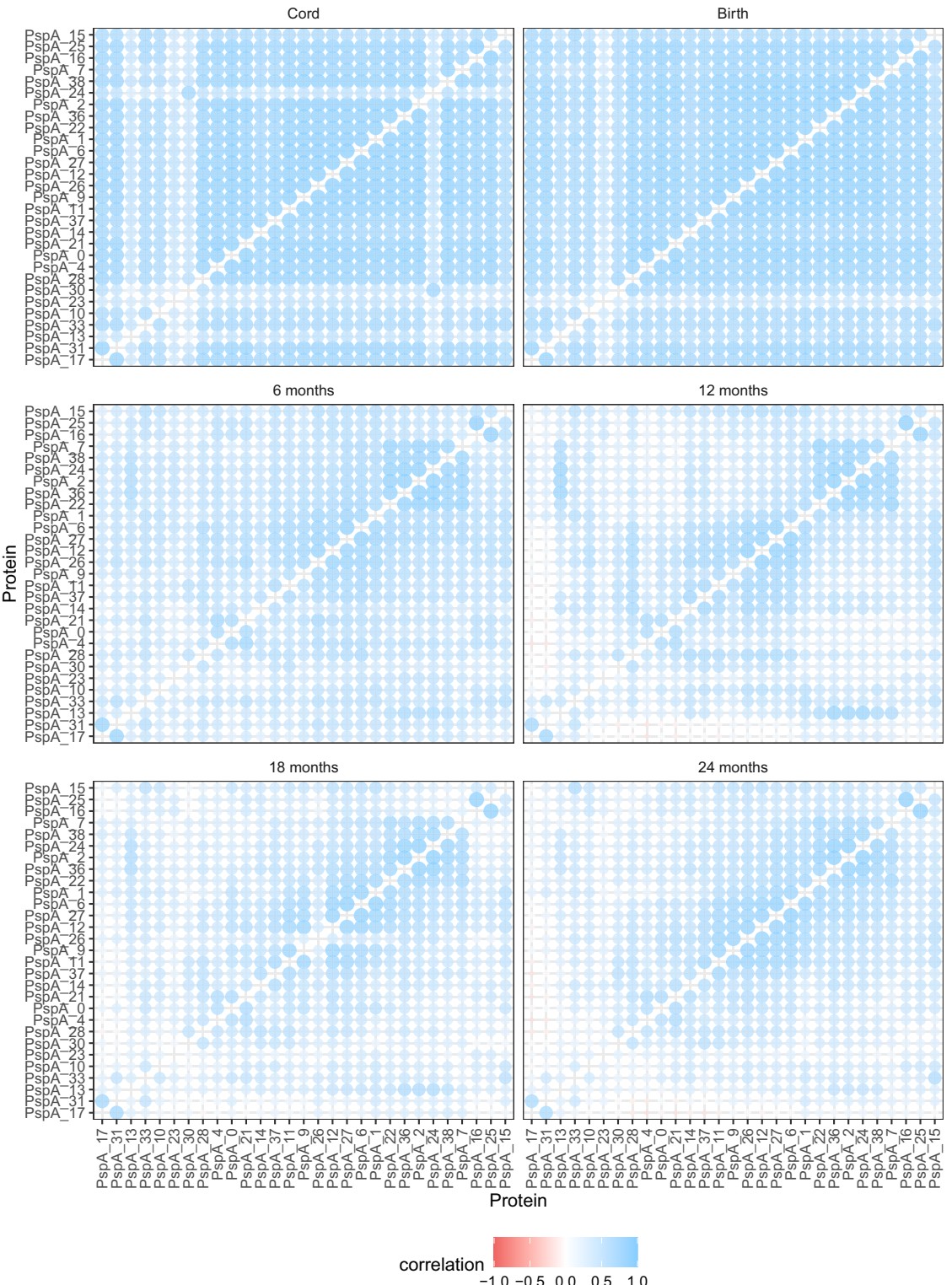

**Fig. 8 | Scatterplot showing the correlation between IgG binding to each pair of PspA variants across individuals at each sampled timepoint.** The colour and size of each point show the strength and direction of the correlation as the Pearson correlation coefficient $R$, calculated across all individuals ($n = 63$). The diagonals of the symmetrical matrices are empty, as these entries correspond to comparisons of proteins with themselves. Each panel shows equivalent data across different timepoints. The variants are arranged by the similarity between their correlation coefficients at 24 mo. Source data are provided as a Source Data file.

differences likely more generally reflect the changing activity and components of the machinery involved in triggering adaptive responses with age[75]. Hence individuals' antibody response is likely determined by the immunogenicity of proteins, their history of exposure, and the maturation of their immune system.

Although the range of proteins targeted by IgG was consistent across individuals, there was quantitative population-wide variation in the strength of individuals' overall anti-pneumococcal response that was determined in the first 6–12 months of life. This was also the period in which there was most evidence of adaptive immunity protecting

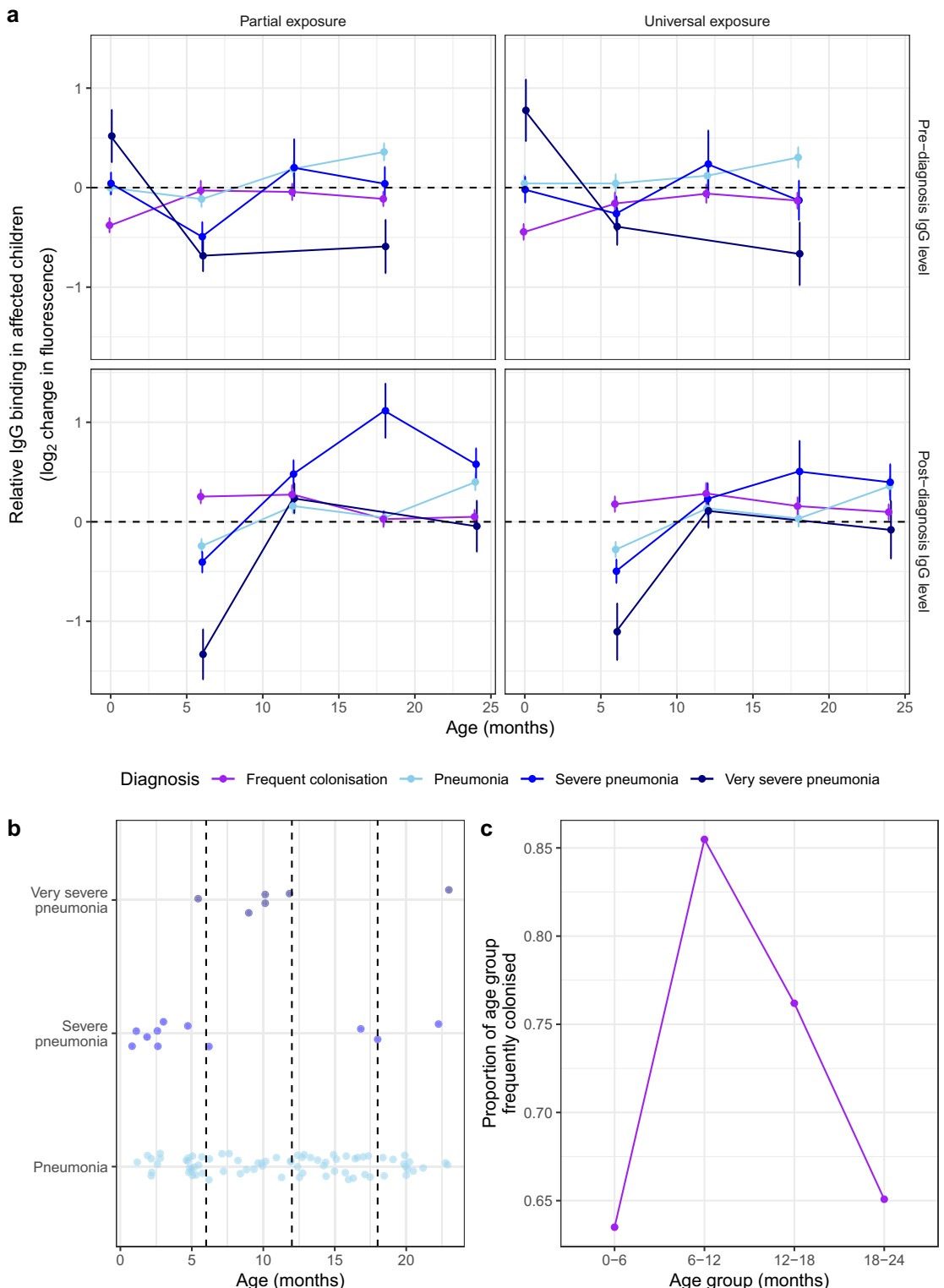

**Fig. 9 | Modelling the antibody-mediated protection against all-cause pneumonia and pneumococcal colonisation. a** Line graph showing the difference in IgG binding in children affected by frequent pneumococcal colonisation, or at least one episode of clinical pneumonia, relative to the rest of the cohort. The points correspond to maximum likelihood estimates from linear mixed effects models (Supplementary Fig. 10). The error bars show the 95% confidence intervals of the estimates. The top row shows the results of models fitted to the IgG binding levels prior to diagnosis of the different conditions; the bottom row shows the results of the same models fitted to the post-diagnosis IgG levels. The left plots show the analysis of proteins to which only a subset of individuals were exposed in the first year of life ($n = 75$). The right plots show estimates from a model of IgG binding to proteins to which all individuals were exposed in the first year of life ($n = 85$). The dashed line represents the null hypothesis of no difference in IgG binding between the conditions. **b** Occurrence of pneumonia diagnoses relative to infant age across the cohort. **c** Colonisation frequency status across the cohort relative to infant ages. Source data are provided as a Source Data file.

against colonisation and pneumonia. This suggests individuals have heightened susceptibility at the nadir of their antibody levels, but that the typical strengths of post-12 mo infant adaptive immune responses are protective. This is consistent with previous associations of lower antibody levels with pneumococcal disease[34,46], and the susceptibility of older children with unusually low antibody levels in this cohort to very severe pneumonia. Hence this study not only demonstrates the ability of children <6 mo to generate endogenous antibodies to pneumococcal proteins, but also highlights the potential benefits of interventions that may stimulate these responses. Furthermore, IgG levels in the mother's blood were positively correlated with those of the infant at 24 mo, suggesting maternal immunisation could both protect neonates and boost responses throughout early childhood[76].

Although antibody levels were associated with a reduced frequency of colonisation in children <6 mo, no such protective effect was evident in older infants, despite the emergence of a strong, polyvalent endogenous response. Hence even if infants' antibodies are effective at preventing symptomatic infection, they do not seem to prevent pneumococcal colonisation of the nasopharyngeal epithelium. A likely mechanism is the modulation of the mucosal immune responses by DCL proteins[37,58,77]. Although they do not preclude the formation of an extensive antibody response, their roles may limit the ability of the host to clear the bacteria following recognition by immunoglobulins, especially of the IgA isotype. The functions by which PspA, PspC and ZmpA achieve this are well-characterised[38,40,58]. This analysis suggests ZmpB also modulates the immune response. Unusually, the highly conserved and immunogenic N terminus contains an LPXTG sortase attachment site, meaning only the first ~75 aa are expected to be attached to the bacterial cell wall[43]. The majority of the protein is instead predicted to be cleaved off and secreted, consistent with it being found at high concentrations in culture supernatants[78–80]. Hence the larger secreted ZmpB fragment appears to be a highly immunogenic, pro-inflammatory protein that is not physically associated with pneumococcal cells, consistent with the protein stimulating TNF-α signalling[43,60,78]. The sequence conservation of the unstructured N terminal region[58] demonstrates its recognition by adaptive immunity is either inconsequential, or more probably beneficial, for ZmpB's function, as the highly variable sequence of the rest of protein suggests the likely catalytic domains of the metalloprotein are under selective pressure to evade immune recognition. This indicates ZmpB's unknown function is directed against antibody responses.

Hence all four DCL likely evolved to facilitate colonisation and transmission through enabling evasion of adaptive immunity in the nasopharynx. However, pneumococcal proteins are unlikely to be under selection to enable avoidance of systemic immune responses, as IPD is an evolutionary "dead end" for pneumococci. Hence the strong polyvalent response to many pneumococcal proteins may be of limited effectiveness in the nasopharynx, enabling repeated carriage episodes of pneumococci sharing conserved epitopes, yet still be active at preventing pneumococci invading the bloodstream. This is perhaps most clearly illustrated by unencapsulated pneumococci, which are effective at colonising the nasopharynx, but very rarely cause IPD[81]. This is consistent with the capsule being the only surface structure that enables evasion of systemic, as well as mucosal, immune responses. Therefore IgA against the majority of pneumococcal proteins may only be effective when other immune mechanisms target, and inhibit, the DCL proteins.

Consistent with this suggestion, previous work has suggested antibodies against a bacterium's PspA and PspC variants are protective against carriage[10,24,26]. This could explain how antibody-based protection against carriage might be genotype-specific[13,16]: variation in DCL, even within strains[51,82,83], means that a newly-acquired pneumococcus may not be affected by adaptive responses targeting previously-carried isolates. Correspondingly, the array demonstrated early infant

IgG repertoires recognised few DCL protein variants compared to the broader immunity observed in umbilical cord and 24 mo samples. Hence the gradual reduction in carriage duration, density and frequency with age[19] could be explained as a consequence of the accumulation of immunity against the range of circulating alleles of the DCL proteins. Furthermore, broad maternal immunity against DCL proteins could also explain both the protective effect of maternal immunity against frequent early colonisation, and the intergenerational correlations in IgG levels, as transplacentally-transmitted antibodies targeting a broad range of DCL proteins may accelerate the production of endogenous responses to other epitopes.

The activity of DCL proteins against all mucosal antibody responses targeting the pneumococcus may also resolve the paradox of differing levels of diversification between this bacterium's antigens[36]: while many ABTs are conserved in sequence across the species, a lack of antibodies recognising a DCL variant may enable the evasion of the broader adaptive immune response. This would explain why the diversity of these loci mirrors the extensive variation in the pneumococcal capsule[6], which also facilitates immune evasion. Yet large antigens (e.g. the pili, PsrP and PclA) may extend too far beyond the bacterial surface for DCL proteins to be effective at limiting their interaction with host immunoglobulins[3]. Hence these adhesins' absence from some genotypes may be a consequence of selection for pneumococci that evade immune recognition through lacking the genes encoding such less essential immunogenic structures. Therefore the DCL proteins are likely a key interface in host-pneumococcus interactions, and a promising target for therapeutic interventions.

Overall, these data highlight the importance of accelerating the generation of anti-pneumococcal antibodies early in life, when individuals are most susceptible to disease. Any such increased production of antibodies is likely to be stable over subsequent years, although longer-term cohort studies will be needed to understand the durability of these elevated responses, and whether they are protective against pneumonia or IPD at a level this analysis was not powered to detect. By highlighting the heterogeneity of individuals' overall immune responses, and the immune evasion mechanisms of pneumococci that have evolved during its adaption to the human nasopharynx, these panproteome data demonstrate the advantages of contextualising antigen-specific responses as components of a polyvalent antibody repertoire.

## Methods
### Study cohort and ethical approval
A cohort of mother-infant pairs in the Maela camp for refugees in Thailand[14,45] was prospectively studied for 24 months from birth. This research complied with all relevant ethical regulations, and was approved by the Ethics Committee of The Faculty of Tropical Medicine, Mahidol University, Thailand (MUTM-2009-306), and by the Oxford Tropical Research Ethics Committee, Oxford University, UK (OXTREC-031-06). All women gave written informed consent to participate in the study. Individuals did not receive monetary compensation for their participation.

The mothers in the cohort were recruited during pregnancy and had a mean age of 26 at birth[69]. Analyses of data from children (n = 63; 34 born female, 29 born male) used samples collected between birth and 24 mo.

### Serum sample collection and panproteome array analysis
At delivery, serum specimens were collected from the mother and the umbilical cord. Serum samples were subsequently taken from infants at monthly intervals up to 24 mo. This study only analysed samples collected at birth, then at 6, 12, 18 and 24 mo. Clinical pneumonia episodes were diagnosed and graded according to WHO criteria[69]. Pneumococcal vaccines were not available to the individuals in the cohort.

The pneumococcal panproteome array was designed to represent all common proteins encoded by the pangenome of pneumococcal isolates collected from nasopharyngeal carriage surveys in Massachusetts[36]. To assay immune responses, serum samples were diluted 1:100 in a 3 mg ml⁻¹ *Escherichia coli* lysate solution, then incubated in Protein Array Blocking Buffer (Maine Manufacturing, Sanford, ME, USA, #10485356) at 16 °C for 30 min. Proteome arrays were incubated in Protein Array Blocking Buffer for 30 min. After removal of the blocking buffer, the arrays were incubated with sera overnight at 4 °C with agitation. After five washes with Tris Buffered Saline (TBS; VWR, Radnor, PA, USA, #J640-4L) containing 0.05% Tween-20 (Thermo Fisher Scientific, Waltham, MA, USA, #J77500-K8), arrays were probed with biotin-conjugated donkey anti-human IgG (Jackson ImmunoResearch, West Grove, PA, USA, #709-065-098) diluted 1:200 in blocking buffer at 16 °C. The arrays were washed three further times with TBS-0.05% Tween 20, then incubated with streptavidin-conjugated SureLight P-3 (Columbia Biosciences, Frederick, MD, USA, #D7-2212) at 16 °C. The arrays were then washed three times with TBS-0.05% Tween 20, three times with TBS, and once with water.

This study analysed the same set of 2343 probes, representing 2190 proteins, used to evaluate responses to a whole cell vaccine[44]. The fluorescence of each probe was measured using a GenePix 4300A High-Resolution Microarray Scanner (Molecular Devices, Sunnyvale, CA, USA) following incubation with the tested sample: either serum from the Maela cohort, or a human monoclonal IgG known to be specific to a pneumococcal capsule polysaccharide. Each of the four polysaccharide-specific monoclonal antibodies was tested at a concentration of 0.5 μg ml⁻¹, as concentrations as low as 0.02 μg ml⁻¹ are typically sufficient to detect binding using the array. Any probe for which the mean fluorescence intensity was below 100 units was regarded as missing data. The level of IgG binding was adjusted by then subtracting the background fluorescence of each probe prior to incubation with serum. The minimum value of this adjusted mean fluorescence intensity was set to one.

The base two logarithms of the values for each probe were then normalised by subtracting the base two logarithm of the median signal intensity of the in vitro transcription and translation control probes on the same chip. This controlled for variation between chips across the study. Hence an IgG binding level of zero corresponded to the background level across the chip; every unit increase above this level corresponded to a doubling of IgG binding specific to the protein.

To identify the most appropriate statistic for summarising IgG responses to the 67 large proteins represented by multiple probes, the correlations were calculated between the IgG measurements from each pair of probes over all ages for each individual. These were summarised as the mean Pearson correlation coefficient, $R$ (Supplementary Fig. 24). The responses to ABT probes were generally more strongly correlated than non-ABT probes (i.e. proteins to which IgG binding was close to background levels), consistent with emerging immune response to antigens contrasting with stochastic variation in the measured response to less immunogenic proteins. The exceptions were the probes to ZmpB proteins, the IgG responses to which correlated less strongly than those to non-ABTs across ages. This was the result of the high levels of IgG binding to the N terminal probes, which potentially saturated the assay's ability to measure antibody responses in some samples. Previously, the most immunogenic probe had been used to represent the response to a protein[36,44], but this risked not accurately describing the dynamics of IgG emergence to ZmpB. However, omitting the N terminal probes would have removed the most immunoreactive epitope in young infants from the dataset. Hence the IgG binding to a protein was calculated as the mean level of IgG binding to the set of probes corresponding to that protein.

Following this calculation of the normalised mean IgG binding to each protein, the analysis was limited to the 1773 proteins defined as immunoreactive, using the criterion that the highest mean IgG binding level was greater than one across the 377 serum samples in the study. Six measurements were missing across the dataset, and were imputed to be the median value recorded from the same probe across all other individuals at the same sampling age. The minimum normalised IgG binding level was set to −2, as any smaller values were still negligible relative to the background level of binding. Therefore all IgG binding values from the array represent a change in the mean fluorescence intensity from the pre-serum background level, adjusted for batch effects, on a base two logarithmic scale.

## Analysis of pneumococcal colonisation

Nasopharyngeal swabs were collected from the mother at birth, and from the infants at monthly intervals while healthy[45]. Additionally, children were swabbed each time they presented with clinical pneumonia. In total, 2718 nasopharyngeal swabs were collected from the 63 mother-child pairs in this cohort, of which 2312 were positive for pneumococci. These isolates were serotyped through latex agglutination[49]. In 79 cases, pneumococcal genotypes were inferred through multi-locus sequence typing, enabling their GPSC to be inferred using previously-established relationships[62].

Previous contemporaneous studies of the Maela camp generated 2758 high-quality single isolate genomes[51–53,61] and 3188 deep sequencing datasets[54]. These were used to identify serotypes that were unambiguously associated with a single GPSC within the camp, which was used to infer the GPSCs of colonising bacteria within the studied cohort for which only serotype information was available.

Using the subset of genomic data that were associated with the 63 children in this study found 357 swabs had been used to generate single isolate genomes; 495 swabs had been characterised by deep sequencing, and 241 swabs had been used to produce both types of data. Many of the remaining 1219 pneumococcus-positive swabs were associated with either only serotype, or only genotype, information. This property was used to impute the missing characteristic using genomic data from the same household where possible, as such cases likely represented continual carriage of a single colonising genotype. This enabled a further 1069 swabs to be linked to sequence data of a representative with a matching GPSC or serotype within the same mother-child pair. A further 18 isolates could be linked to a genome of the same serotype or GPSC within another mother-child pair in this cohort. Finally, 16 isolates were represented by genomic data arising from other individuals within the Maela camp using the same criteria. Hence the gene content of 2196 swabs could be inferred from 1117 sequence datasets. This left only 116 instances of pneumococcal colonisation for which no representative genomic data were available.

## Linking genomic and proteomic data

Each of the 1117 genomic datasets was assembled using Velvet[84]. To annotate these contigs consistently with the original dataset used to design the panproteome array[35], protein coding sequences were inferred from the consensus of the annotations generated by Glimmer[85] v3.0.2 and Prodigal[86] v2.6.3. This generated a database of 2,427,067 proteins, from which a non-redundant set of 202,057 unique sequences could be extracted. All pairwise comparisons between the set of non-redundant protein sequences were calculated using BLASTP[87] v2.7.1, both with and without filtering of low-complexity regions with segmasker[87] v1.0. The same approach was used to align these proteins with the original dataset used to design the array[35]. COGcognitor[88] was used to process these BLASTP alignments to integrate the new data into the existing protein clustering; each new protein was only assigned to a single, best-matching cluster. Proteins that were not assigned to a cluster by this approach were matched through pairing with a protein in the original dataset if the E value of the match was both below 0.05 following a Bonferroni correction for the number of comparisons analysed, and at least 10⁴-fold lower than the match to the next most similar protein cluster.

The DCL were categorised at a higher-level resolution, and therefore could not be assigned to isolates through this clustering process[36]. Instead, the inference of PspA and PspC variants used sequence read mapping, and the inference of ZmpA and ZmpB variants used an amino acid *k*-mer based approach, as described previously[36]. The minimum threshold scores for inferring the presence of an allele were identified from empirical distributions generated specifically for this study, to reflect the differences in sequence coverage and read length in this analysis. Only one variant of each DCL was assigned to single isolate genomes, whereas the maximum number of variants that could be assigned to deep sequencing datasets reflected the number of genotypes present, defined as the count of unique combinations of GPSC and serotype previously identified within the data[54]. Functional domains were identified within DCL proteins using the NCBI CD-Search tool[89]. To analyse the conservation of the ZmpB N terminal region, the full-length protein sequences were aligned with Muscle[90], and displayed with ggmsa[91].

Four additional proteins, which were not part of the original genome clustering[36], were identified individually. The presence of the long, repetitive protein PsrP was inferred from the tightly-linked SecA2 protein, as described previously[36]. The distributions of PblB and ZmpE were inferred through querying the 1117 draft genome assemblies with the respective protein sequences using TBLASTN (Supplementary Fig. 34). The distribution of the phage amidase was inferred from the distribution of prophage, determined through querying the draft assemblies with BLASTN using a database of 61 viruses generated through combining previous studies of pneumococcal prophage diversity[82,92,93] (Supplementary Fig. 34). The cellular lytic amidase, the choline-binding domain, and all penicillin-binding protein variants were assumed to be ubiquitous across all isolates.

The frequencies of all protein clusters across the population were calculated. The 1183 proteins present in more than 99% of genomes were considered core proteins. These proteins were assumed to be expressed by the pneumococci detected in the 116 swabs for which no genomic data were available. The classification of proteins into ABTs and non-ABTs, and into functional categories, used that published previously[94].

### Statistics and reproducibility
Statistical analysis was undertaken in R[95,96]. No statistical method was used to predetermine sample size. One serum sample was not analysed, due to a failure to generate data of sufficient quality. No other data were excluded from the analyses. There was no randomised allocation of individuals to groups, and no blinding to associations between variables during the analysis.

### Sample clustering, embedding, correlation and modelling
Unsupervised clustering of the IgG binding data used SIMLR[97]. The number of clusters was ascertained by using the recommended heuristics to compare divisions of the dataset into up to ten clusters. The optimal number of clusters was found to be six. Hence the data were split into this number of categories using a tuning parameter of 12, which increased the robustness of the clustering to changes to the input data and other parameter values.

Univariate testing for associations between immune clusters and discrete environmental and physiological variables used Fisher's exact tests of two-by-two contingency tables. These were generated by converting discrete variables into a binary classification of whether or not individuals belonged to the modal group, and comparing this categorisation with the immune clustering of individuals at 24 mo.

Analysis of genetic and immunological variation with t-distributed stochastic neighbour embedding used the R package tsne[98]. These analyses used a perplexity of 40, and were fitted with a maximum of 1000 iterations, unless otherwise specified. Complementary UMAP analyses used the R package umap[99], with a default perplexity of 40, fitted with a maximum of 200 iterations.

The analysis of cross-immunity between DCL variants used the R package corrr[100] to calculate the Pearson correlation in the IgG response to pairs of proteins across individuals at a specified sampling age. The correlation matrix was ordered using the optimal leaf ordering method implemented in the R package seriation[101] using the values calculated at 24 mo.

The fitting of linear mixed effects models used the R package lme4[102]. Additional analysis and visualisation of these data used the tidyverse[100], ggpubr[103] and cowplot[104] packages.

### Reporting summary
Further information on research design is available in the Nature Portfolio Reporting Summary linked to this article.

## Data availability
The raw data are available from a Github repository (https://github.com/nickjcroucher/maelaAntigens), archived within Zenodo (https://doi.org/10.5281/zenodo.10333337). The processed immunological, epidemiological and genetic data are available from FigShare (https://doi.org/10.6084/m9.figshare.23755524.v2). Source data for all figures are also available from FigShare (https://doi.org/10.6084/m9.figshare.24781587.v1). The raw sequence data used in this study are available from the European Nucleotide Archive with the accession codes listed in Supplementary Data 2.

## Code availability
The code used in this analysis is available from https://github.com/nickjcroucher/maelaAntigens. The version used in the described analysis has been stably archived (https://doi.org/10.5281/zenodo.10333337).

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

## Acknowledgements

Research reported in this publication was supported by The Bill & Melinda Gates Foundation under a contract to Antigen Discovery, Inc. (contract #24575, WO#4). A Sir Henry Dale fellowship jointly funded by Wellcome and the Royal Society (grant 104169/Z/14/A), and the UK Medical Research Council and Department for International Development (grant MR/R015600/1), supported N.J.C. Wellcome (grant 206194) supported S.D.B. and (grant 083735/Z/07/Z) P.T. The funders had no role in study design, data collection and analysis, decision to publish, or preparation of the manuscript.

## Author contributions

D.G., P.T. and J.J.C. conceptualised and designed the study. S.D.B. organised the collection of genomic data. P.T., C.T. and F.N. organised the collection of epidemiological and clinical data. D.G., P.T., J.J.C. and X.L. organised the collection of the immunological data. J.J.C., T.Q.L., J.V.P., C.H. and A.A.T. generated the immunological data. N.J.C. and J.J.C. analysed the immunological data. N.J.C. and P.T. analysed the epidemiological data. N.J.C. analysed the genomic data. N.J.C. drafted the manuscript, which was edited and approved by all authors.

## Competing interests

J.J.C., T.Q.L., J.V.P., C.H., A.A.T. and X.L. are employees of Antigen Discovery, Inc. X.L. has an equity interest in Antigen Discovery, Inc. N.J.C. has consulted for Antigen Discovery Inc. and Pfizer, has been invited to attend meetings organised by Merck, and has received an investigator-initiated award from GlaxoSmithKline. S.D.B. has consulted for Pfizer and Merck. The remaining authors declare no competing interests.
