## [Peer Review File · Nature Communications]

REVIEWER COMMENTS

Reviewer #1 (Remarks to the Author):

This is a very in depth and comprehensive manuscript on the antibody response to pneumococcal exposure in infants. I have no major concerns, but please address the following:

page 11 going into 12 - they bring up "seven antigens that triggered a doubling of antibody responses" but then don't elaborate on that concept at all and don't mention any of the seven antigens.

Are variants of the different serotypes here? Please add some elaboration on the different variants and different alleles between antigens - like PspA 17 vs PspA 23

Many r squared values are 0.25 and below? Please clarify if these are supposed to show significance?

Reviewer #2 (Remarks to the Author):

The authors have amassed an impressive set of data on antibody concentrations, colonization patterns, and bacterial genomics. Unfortunately, this massive amount of data is presented in a way that is not easy to digest. There is way too much information presented. Without a clear structure or goal, it is very difficult to follow the thread and identify key findings. The paper is top heavy, with 56 supplementary figures, and a number of sections of the results that are not clearly connected. There is surely important information in here, but it needs a major overhaul in terms of presentation. Here are a few specific comments:

1) The results section is particularly difficult to get through. It has a mix of results, methods, and interpretation, which breaks up the flow and makes it difficult to follow the thread.

2) It seems the antibody kinetics vs pneumonia analysis is backwards for the question the authors are trying to answer. Ideally, they would evaluate antibody levels before the pneumonia episode as a predictor of risk. Evaluating antibody levels after the fact are hard to interpret.

3) ““These weakened, yet still positive, correlations suggest higher maternal antibody levels are associated with elevated infant IgG responses in the first years of life.” Could these patterns also reflect variations in exposure between households, which would induce a correlation in antibody levels between mother-infant pairs, unrelated to antibody transfer or waning

4) I'm not sure this statement/analysis makes sense: “After correcting for previous exposure to ABTs, the age-associated increase in IgG binding was relatively weak across non-DCL proteins” How would responses increase if not due to previous exposure?

Reviewer #3 (Remarks to the Author):

This manuscript by Croucher and colleagues investigated the development of IgG to pneumococcal proteins in infants during the first 2 years of life. They utilised a novel panproteome approach to detect responses to 2,190 proteins in serum samples. They assessed the relationship of key pneumococcal protein responses with carriage episodes and during pneumonia (using phenotypic and genomic approaches) to provide insights into the role of the antibody responses in protection. The study was conducted in mother-infant pairs living in a refugee camp in Thailand where there was no PCV use. Longitudinal samples were measured from birth, including at 6, 12, 18 and 24 months in 63 infants. Key findings from this study revealed that IgG levels were highest at birth (from maternal antibody), but that an infant's immune response to pneumococcal proteins became established by 12m of age. In addition, they found that higher IgG was associated with reduced pneumonia incidence under 6 months of age while carriage and pneumonia was able to stimulate pneumococcal protein IgG levels in older adults using both assay data and modelling approaches.

This is an interesting and important set of data that employs a combination of proteomic and genomic studies to better understand infants acquired immunity to pneumococcus in a high burden setting. It is well written and provides food for thought for ongoing studies in this area.

A few points that the authors may wish to consider:

1. Was clinical pneumonia confirmed to be pneumococcus or was pneumococcal status based on the NP swab result?
2. Have the authors analysed the IgG data taking into account the time of carriage and/or pneumonia. My understanding is that this was done in the 6 months prior but was there any association with seroconversion and time of episode?

3. How does the protein-specific IgG levels correlate with serotype-specific IgG responses (capsule-specific) and do they show similar kinetics over time and with exposure? Is one a better predictor over the other?
4. The authors mention that the response is likely to be a reflection of total IgG but was this measured?
5. If protein IgG responses do not block colonisation, then how would this fit into the larger picture of how to protect children against pneumococcal infections? As the authors rightly state, protein-based vaccines have had limited success.
6. The authors state that ZmpB binds to the invariant region of IgG – would this suggest some Fc-mediated effector functions? Has/can this be measured? Some discussion of this might be worthwhile.

Reviewer #4 (Remarks to the Author):

The authors studied adaptive immune responses to pneumococcus in a prospectively-sampled cohort of 63 Maela infants during the first 2 years after birth, using integrated analyses of longitudinal panproteome array and pneumococcal genome data. With several dozens of Supplementary Figures, they extensively analyzed the valuable data and deeply assessed their results including detail parts. This study would well support the usefulness of the panproteome profiling in identifying heterogenous adaptive immune responses to pneumococcus variants. At the same time, their extensive presentations and descriptions likely make readers sidetracked from the main story. When considering Nature Communications is a general scientific journal, the current version of manuscript should be revised for its readership.

Major points:

1. The authors concluded that “zinc metalloprotease B, exhibited characteristics of a high-affinity immunoglobulin-binding protein” (lines 18–19 of page2; and similar sentences in lines 7–8 of page 17). In my understanding from line 14 of page 5 to line 24 of page 6, this conclusion was derived mainly from Fig. S1B, which demonstrates that ZmpB N-terminal probes predominantly exhibited the maximum IgG binding among all probes. However, as the authors mention (lines 4–5 of page 6), N-terminus region of ZmpB is little exposed to the surface and thus unlikely functional as a physiological antigen within our body. Hence, there is a gap to conclude physiological function of ZmpB from their results. In addition, although they describe this conclusion even in Abstract, I do not think that this conclusion is an important finding in this study. Indeed, compared to the other diverse core loci (DCL) proteins, “the function of ZmpB remains unknown” (lines 19–20 of page 4). Nevertheless, the immunoglobulin-binding function could be easily assumed for ZmpB. I acknowledge that their demonstration of actual evidence is important in science, but this level of finding is not regarded as a significant advancement in modern Nature Communications. Therefore, they should remove or tone down this point.

2. The authors unexpectedly observed that the responses to ZmpB probes were less strongly correlated with age than non-antibody-binding target (non-ABT) probes (Fig. S1A), and interpreted that it was likely because of “saturating levels of antibody binding” (line 34 of page 5). I totally agree with their logical explanations. However, their interpretation also indicates that the signals of these saturated probes (in their measurement condition) did not meet the sufficient data quality for assessment. Therefore, they should eliminate these probes during data processing and analyze the data without these probes throughout this study. Related to the above major point 1, it might be good to remove N-terminal probes for the other DCL proteins as well, depending on their choice of criteria. Of course, they must explicitly describe this kind of data exclusion in the Method section and Nature journal’s Reporting Summary, but the data exclusion itself is reasonable. (Even after updating the results, it would be fair that they include the current Fig. S1–3 as supplementary information.)

3. The authors observed “substantial decline in IgG binding to all ABT types except ZmpB” (lines 31–32 of page 7) between birth and 12 months, followed by gradual rises of all non-ZmpB ABTs (Fig. 2A). What is a potential reason of this ZmpB exception? Again, if it was due to the N-terminal probes of ZmpB, they should have presented the result without these probes.

4. The authors should adjust all available potential confounding factors (as much as possible) within statistical tests/models throughout this study, especially because this study investigated only one cohort. For example, they addressed the possibility of several potential confounding factors for the relationships between IgG binding and ABTs per the factor (lines 8–27 of page 15; Figs. S52–56). However, it is better to present one single figure (Fig. 4) with these factors adjusted in statistical tests. Similarly, all linear mixed models they tried did not include the terms about these potential confounding factors, but they should have added these factors as fixed effects (without interactions). If the small sample size was the reason why they did not consider confounding factors within statistical tests/models, alternative method would be regressing out the potential confounding factors during data preprocessing and then using the residuals as signals throughout this study.

Minor points:

1. The authors frequently use “reproduced/reproducible across the cohort” (e.g., line 10 of page 2; line 1 of page 8). However, in life science, ‘reproduce a result’ usually indicates that the result is obtained from independent experiments/observations after the first experiment/observation. Hence, it would be better to use ‘consistently/commonly observed among the individuals’ instead, for example.

2. “suggesting all four DCL modulate the host mucosal immune response” (lines 19–20 of page 2): the former part of this sentence says about only one DCL (ZmpB), and thus, until reading the Introduction section, readers cannot understand why something about all four DCL proteins can be suggested from the former part.

3. “Fig. S1” (line 23 and line 33 of page 5): the authors should exactly cite ‘Fig. S1A’ and ‘Fig. S1B’, respectively.

4. “Fig. S1” (line 3 of page 6) would be a typo of ‘Fig. 1A’.

5. “which shares structural similarities with PspC (Fig. S4)” (lines 16–17 of page 6): this way of figure citation is as if Fig. S4 shows structural (e.g., amino acid sequence-level) similarities between the ZmpB N-terminus and PspC. The authors should improve this sentence and citation position.

6. “Hence the IgG binding for ... in the subsequent analyses.” (lines 19–24 of page 6): these sentences are about not logical consequences but just the criteria that the authors chose to use in this study. Therefore, these sentences are more suitable for the Method section. Basically, the Results subsection “ZmpB is a high-affinity antibody-binding protein” (line 14 of page 5 to line 24 of page 6) includes many supplementary information related to measurement/analysis conditions. Of course, all of them must be considered during the progress of study (and so, I acknowledge the authors’ sincerity for science). However, the manuscript is not a notebook but a scientific article. Hence, the authors could simplify this subsection or move the details to Supplementary Note while citing the Method section.

7. “Comparisons of the 377 IgG datasets identified extensive cross-sectional and longitudinal heterogeneity (Fig. S5, S6, S7)” (lines 27–28 of page 6): Figs. S5–7 are not good presentations to claim heterogeneity among the individuals. These figures are negative results in point of clustering individuals, but it is possible that it was just due to a bad projection from data space. (I believe the authors checked several parameters for the projection, though.) Basically, I do not think that Figs. S5–7 satisfy the roles of Supplementary Information for the findings of this study. The authors could remove these ‘negative’ results from the manuscript and deposit them to FigShare (lines 9–10 of page 24) as an original result.

8. “approximately half” (line 5 of page 7): it seems overestimated. Probably, were IgG binding values log₂-scaled? Then, the authors should add ‘log₂-scale’ to Y-axis title in all IgG binding figures.

9. “Fig. S4” (line 6 of page 7) would be a typo of ‘Fig. S10’.

10. “Consistent with individuals ... (Fig. 1C). Hence ... the child.” (lines 15–18 of page 7): these sentences have no logical ‘hence’ relationship. The pairs of umbilical cord and birth samples could be observed even if the unique property was derived from not the mother but the child. So, the authors should describe the latter sentence as a possible explanation.

11. “Those belonging to the infant high cluster ... datasets (Fig. S11)” (lines 21–24 of page 7): the infant high and low are inverse between the sentence and Fig. S11. The authors should check the sentence or color coding of Fig. S11.

12. It is difficult to recognize the median line and violin shape in some violin plots (e.g., Fig. 1A). They should improve these plots, for example, by changing colors and placing the median line and violin to the top layer, like Fig. 4.

13. “Pearson correlation coefficient with umbilical cord sample” (Y-axis of Fig. 3B) would be a typo of ‘Pearson correlation coefficient with 6-months sample’.

14. Figure 6 and Figs. S41–43: the authors could improve these figures, because half parts of these large figures are redundant, because information about row and column clustering is removed, and because the difference of values are not easily recognizable. Probably, they could summarize the correlation value distributions per variant with violin plot (with statistical tests for the distributions).

15. “shows” (line 4 of page 43) → ‘show’

16. Fig. S1B: because many points overlap with each other, it would be better to describe how many ZmpB N-terminal probes actually exhibited the maximum signals among all probes.

NCOMMS-23-35488: Response to Reviewer Comments

All reviewer comments are reproduced in **boldface**. Quotations from the manuscript are presented in **blue**. As the journal requires a reduction in the word count of the article, changes have been made to shorten the main text, in addition to those made in response to reviewers' comments.

Reviewer #1:

This is a very in depth and comprehensive manuscript on the antibody response to pneumococcal exposure in infants.

We thank the reviewer for their positive comments on the manuscript.

I have no major concerns, but please address the following:

page 11 going into 12 - they bring up "seven antigens that triggered a doubling of antibody responses" but then don't elaborate on that concept at all and don't mention any of the seven antigens.

We thank the reviewer for highlighting the disconnect between different sections of the analysis, as presented in the original version of the study. The seven antigens referenced were those labelled in Fig. 5A, and discussed in the subsequent section of the text: PitB, ZmpC, ZmpD, PspA_15, PspA_17, PspA_23 and PspC_32. However, we did not make this connection clear. By improving the processing of the model output, we have now identified an eighth protein, NanC. We have now edited this section of the manuscript to link these analyses together:

Eight ABTs triggered a doubling, or more, of the relevant antibody response after exposure to the ABT (Fig. 6a). These included four non-DCL rare antigens: the type 2 pilus protein PitB⁶⁶ (CLS02871; present in 18.5% of pneumococci); the neuraminidase NanC (CLS01160; present in 16.8% of pneumococci)³⁶; ZmpC⁶⁷ (CLS01991; present in 5.77% of pneumococci) and ZmpD³⁹ (CLS02608; present in 10.0% of pneumococci).

Are variants of the different serotypes here? Please add some elaboration on the different variants and different alleles between antigens - like PspA 17 vs PspA 23

The pneumococcal population isolated in the Maela cohort expresses a broad range of serotypes. We attempted to summarise this in Fig. S18 in the original version of the manuscript (now Supplementary Figure 30), which shows the variety of serotypes recovered in the context of the dates of pneumococcal isolation from each individual. However, we appreciate this does not enable the reader to interpret the overall distribution of variable antigens across the population. We have therefore added a new supplementary figure (Supplementary Figure 32) that shows the distribution of serotypes across the GPSCs that comprise the population, enabling a direct comparison with the distribution of the DCL protein variants, shown in Supplementary Figures 39-42. Additionally, it is now stated in the text:

These isolates represented 89 different Global Pneumococcal Sequence Clusters (GPSCs)⁶² and 56 different serotypes (Supplementary Fig. 32).

We thank the reviewer for their helpful suggestion to describe the variation between PspA sequences. The diversity of the variants of the DCL proteins has previously been phylogenetically analysed in Croucher *et al* (2017; *PNAS* **114**(3):E357-366). This analysis provides helpful context on the responses to the PspA variants described in the main text. In particular, the limited cross-immunity between PspA 23 and other variants is likely a consequence of the divergence of this sequence from the other PspA representatives. Similarly, PspA 17 is very similar to PspA 31, which was the only other variant to which it exhibited any substantial cross-immunity in older infants. We have therefore now edited the manuscript to state:

The PspA proteins to which there was a detected relationship between exposure and response (Fig. 6) seemed likely to represent examples relatively unaffected by cross-immunity elicited by other variants. This was not clearly the case for PspA 15, but responses to PspA 23 correlated only weakly with those to other PspA variants, even in the umbilical cord and maternal birth samples (Fig. 8). This may be a consequence of PspA 23's divergent sequence relative to other PspA variants³⁶. Similarly, immunity to PspA 17 in older infants only correlated strongly with the highly-similar PspA 31³⁶. Hence the accumulation of antibody binding to such variants (Fig. 6b), which are minimally affected by cross-immunity, should provide a reliable measure of the immune response to individual variants of this class of diverse proteins.

Many *r* squared values are 0.25 and below? Please clarify if these are supposed to show significance?

We thank the reviewer for highlighting the lack of clarity regarding the significance of relationships illustrated with scatterplots. We have now added Pearson *R* values, and the associated *p* values, to all scatterplots. In all cases except for Fig. 4, the correlations were found to be highly significant.

Fig. 4 shows the correlation between the IgG binding levels in the early and late serum samples for each protein across the individuals in the cohort, grouped by ABT type. The Pearson correlation coefficient, *R*, for many of these was below 0.25, and often close to 0.1. Therefore the individual relationships between IgG level measurements across individuals are not typically significant; we have added a supplementary figure (Supplementary Figure 29) that shows the relationship at the level of individual antigens that exhibit *R* values close to 0.25 (CLS01620) and 0.1 (CLS01065). Therefore studies of small numbers of antigens would not necessarily detect such a relationship, unless they had a much larger cohort size. The advantage of the array is being able to analyse hundreds of proteins in parallel. This demonstrates that the distribution of *R* values significantly deviates from the null expectation of zero across almost all ABT types at almost every timepoint, as is now illustrated on Fig. 4 by Wilcoxon rank sum tests. We have now clarified this in the text:

Hence the association between maternal and infant IgG responses is weak at the resolution of an individual antigen (Supplementary Fig. 29), but combining associations across many ABTs demonstrates higher maternal antibody levels are associated with elevated infant IgG responses in the first years of life.

The use of R values, rather than R^2 values, is necessary to distinguish positive and negative correlations, as R^2 values are always positive. We have clarified some erroneous use of the R^2 notation in the main text.

Reviewer #2:

The authors have amassed an impressive set of data on antibody concentrations, colonization patterns, and bacterial genomics.

We are grateful to the reviewer for their evaluation of the manuscript.

Unfortunately, this massive amount of data is presented in a way that is not easy to digest. There is way too much information presented. Without a clear structure or goal, it is very difficult to follow the thread and identify key findings. The paper is top heavy, with 56 supplementary figures, and a number of sections of the results that are not clearly connected. There is surely important information in here, but it needs a major overhaul in terms of presentation. Here are a few specific comments:

We appreciate this is a complex analysis, and we have sought to clarify the presentation of the data in response to the reviewers' feedback.

1) The results section is particularly difficult to get through. It has a mix of results, methods, and interpretation, which breaks up the flow and makes it difficult to follow the thread.

We have now extensively revised the Results section as suggested. We have moved some of the technical detail of the statistical analysis, which was unlikely to be of broad interest to readers, to the Methods section. We have also removed sections that focussed on data interpretation (e.g. inferring the function of ZmpB, and proposing the likely immune mechanisms causing the poor response to some adhesins) from the Results, instead summarising these parts in the Discussion.

2) It seems the antibody kinetics vs pneumonia analysis is backwards for the question the authors are trying to answer. Ideally, they would evaluate antibody levels before the pneumonia episode as a predictor of risk. Evaluating antibody levels after the fact are hard to interpret.

This is a very good point. We have now revised the analysis of the relationship between IgG and pneumonia to include both antibody levels before, and after, cases of pneumonia, shown in a revised Fig. 9. Overall, the synthesis of the pre-pneumonia and post-pneumonia IgG concentration modelling suggests that disease of all severities is associated with low IgG levels at the nadir of immunity at 6 mo,

whereas very severe pneumonia is associated with low anti-pneumococcal antibodies prior to disease onset at older ages as well. This suggests anti-pneumococcal antibodies are protective against such infections. Yet no such signal of differential IgG levels is observed after cases of very severe pneumonia, suggesting that instances of the disease are associated with the accumulation of increased anti-pneumococcal antibodies.

Surprisingly, we observed that very severe pneumonia early in life was associated with higher levels of maternal antibodies. However, we have also added a panel to Fig. 9 that shows the timing of cases of pneumonia, revealing most disease occurs close to 6 mo, when the influence of maternal antibodies has waned.

We now summarise this in the main text as:

At 6 mo, lower anti-pneumococcal IgG levels were associated with increasing severity of pneumonia in the prior, and following, six months. However, umbilical cord IgG levels were not associated with protection in early life. This likely reflects both the clustering of cases close to 6 mo (Fig. 9b), at the nadir of IgG levels, and the diverse aetiology of pneumonia. By contrast, frequent colonisation in the first six months of life was associated with lower IgG levels in the umbilical cord sample, but higher IgG levels at 6 mo. Colonisation frequency ceased to be associated with differential IgG levels as the endogenous response strengthened with age (Fig. 9c). Analogously, very severe pneumonia was associated with low IgG levels in the preceding serum sample at 6 and 18 mo, but these antibody levels rose to normal levels post-disease. This suggests low antibody levels, particularly common at 6 mo, increase susceptibility to pneumococcal colonisation and pneumonia, both of which are sufficiently immunostimulatory to elevate responses towards the population-wide average.

3) “These weakened, yet still positive, correlations suggest higher maternal antibody levels are associated with elevated infant IgG responses in the first years of life.” Could these patterns also reflect variations in exposure between households, which would induce a correlation in antibody levels between mother-infant pairs, unrelated to antibody transfer or waning

This is an interesting point that we sought to address with Fig. S31 in the original submission (now Supplementary Figure 44). This demonstrated that there was no strong association between the strength of an infant’s overall IgG response to pneumococci and the timing of their first pneumococcal colonisation (panel a), or the frequency with which they were colonised by pneumococci (panel c). Hence we did not find any evidence of variation in exposure contributing to the distinction between infant immune clusters.

In the linear mixed effects modelling analysis of colonisation and pneumonia, there was evidence that frequently-colonised children have higher IgG responses at 6 mo (Fig. 9), but this relationship was not observed at other sampling times (as described in the text from the manuscript above). Hence differences in the frequency of colonisation did not appear to cause the stable differences in IgG levels across ages. As we did not have information on the frequency of colonisation of the mother prior to

birth, we were not able to explore the joint effects of colonisation frequency on IgG responses across both family members.

4) I'm not sure this statement/analysis makes sense: "After correcting for previous exposure to ABTs, the age-associated increase in IgG binding was relatively weak across non-DCL proteins" How would responses increase if not due to previous exposure?

This test was intended to understand what drove the age-associated increases in the IgG response: either repeated exposure to antigens, or the maturation of the infant immune system with age, independent of any exposures. We appreciate this hypothesis was not clear in the original version of the manuscript.

To better understand the primary driver of increases in IgG levels with age, we have now added a main figure (Figure 7) that analyses the changes in antibody binding with age. We hypothesised that the largest increases in IgG binding would represent boosting of an individual's relatively weak IgG responses. Therefore we compared the changes in IgG binding to each antigen in each individual between samples, representing the strengthening of their immune response, with a measure of the relative weakness of the same response at the previous timepoint, which corresponded to the difference between the individual's response to a protein, and the population-wide mean response to the protein. This identified highly significant negative correlations across responses to all protein types at all sampled ages. Hence the largest increases in IgG binding with age were observed to proteins to which the individual previously had a relatively weak response. This is consistent with our hypothesis of re-exposure causing boosting of weak responses, but not with a uniform increase in IgG binding that might be driven by exposure-independent immune maturation. Therefore we have now edited the text to state:

Yet the model fits suggested mean IgG responses increased across ABT types from 12 mo onwards (Supplementary Fig. 53). This rise could represent exposure-independent effects of immune system maturation, or boosting of responses by re-exposure. To distinguish these hypotheses, the change in IgG binding to a specific ABT in an individual between successive serum samples was compared with the difference between the individual's response, and the population-wide mean response, to the same ABT at the earlier sampling time (Fig. 7). The negative correlation between these quantities across ABT types and ages demonstrates the largest rises in IgG levels against an ABT occur in individuals previously exhibiting a low response to that antigen, relative to the population-wide mean. This is consistent with repeated encounters with pneumococci stimulating increases in any poor responses to immunogenic proteins, but not with a broad strengthening of all antibody responses with age. This is compatible with the stability in the median IgG levels to each ABT (Fig. 6b). Hence the gradual increase in population-wide adaptive immunity with age is driven by boosting of individuals' weakest responses to antigens following re-exposure.

Reviewer #3:

This manuscript by Croucher and colleagues investigated the development of IgG to pneumococcal proteins in infants during the first 2 years of life. They utilised a novel panproteome approach to detect responses to 2,190 proteins in serum samples. They assessed the relationship of key pneumococcal protein responses with carriage episodes and during pneumonia (using phenotypic and genomic approaches) to provide insights into the role of the antibody responses in protection. The study was conducted in mother-infant pairs living in a refugee camp in Thailand where there was no PCV use. Longitudinal samples were measured from birth, including at 6, 12, 18 and 24 months in 63 infants. Key findings from this study revealed that IgG levels were highest at birth (from maternal antibody), but that an infant's immune response to pneumococcal proteins became established by 12m of age. In addition, they found that higher IgG was associated with reduced pneumonia incidence under 6 months of age while carriage and pneumonia was able to stimulate pneumococcal protein IgG levels in older adults using both assay data and modelling approaches.

This is an interesting and important set of data that employs a combination of proteomic and genomic studies to better understand infants acquired immunity to pneumococcus in a high burden setting. It is well written and provides food for thought for ongoing studies in this area.

We thank the reviewer for their detailed evaluation of the work.

A few points that the authors may wish to consider:

1. Was clinical pneumonia confirmed to be pneumococcus or was pneumococcal status based on the NP swab result?

Clinical pneumonia was diagnosed symptomatically, according to WHO recommendations, and the aetiological agent was not established. This is because identifying the causal pathogen is extremely challenging, given the range of viruses and bacteria that can cause pneumonia, and the existence of a lung microbiota even in healthy patients. Hence the pneumococcal colonisation status was inferred from nasopharyngeal swabbing at the time the infant presented with symptoms. Such data is subject to the caveats that a positive swab does not mean pneumococci are the causative agent of the pneumonia, and a negative swab is not conclusive evidence that there is no pneumococcal infection of the lungs. We have now clarified the proportion of pneumonia swabs that were positive for pneumococci in the Results:

The accumulation of IgG responses enabled testing of whether they protected against all-cause clinical pneumonia. Across the cohort, nasopharyngeal samples were collected during 158 episodes of non-severe pneumonia (93% positive for pneumococci), 16 episodes of severe pneumonia (69% positive for pneumococci), and 10 episodes of very severe pneumonia (90% positive for pneumococci)⁶⁹.

The Methods section now states:

Clinical pneumonia episodes were diagnosed and graded according to WHO criteria⁶⁹.

2. Have the authors analysed the IgG data taking into account the time of carriage and/or pneumonia. My understanding is that this was done in the 6 months prior but was there any association with seroconversion and time of episode?

We thank the reviewer for this interesting question. In the original submission, we plotted the data for the differences in IgG response following exposure to an antigen in the first month of life, first six months of life, the six months before serum sampling, and the month before sampling (previously Fig. S32-S35; now Supplementary Figures 45-48), or any time since birth (previously Fig. 4; now Fig. 5). These results did not suggest the timing of exposure was important in determining the strength of the IgG response. These were tested using linear mixed effects models to infer whether the IgG response was primarily determined by exposure in the first six months of life, the six months prior to serum sampling, or at any time since birth (previously Table S6; now Supplementary Table 7). This demonstrated that the timing of exposure did not seem to matter for the IgG response. We have now summarised this interpretation as:

Hence individuals maintain stable IgG repertoires recognising their earliest colonisation episodes, which expand intermittently throughout life as they encounter novel proteins.

We have now also improved the analysis of IgG levels relative to the timing of the pneumonia episodes. This helpful suggestion was also made by Reviewer #2, and consequently we have added an analysis of the IgG levels prior to episodes of pneumonia, as well as afterwards (Fig. 9). This revealed pneumonia of all severities in the first six months of life was not associated with low maternal antibody levels, but was associated with low post-disease IgG in the serum sample taken at 6 mo. This highlighted the sensitivity of the timing of disease relative to the IgG measurements, as the incidence of non-severe and very severe pneumonia was highest shortly before the 6 mo sample was taken, suggesting this sample is more representative of the individual's immune status at the time of disease than the sample collected at birth. Consequently, we have added a panel to Fig. 9 showing the ages at which clinical pneumonia episodes occurred in the cohort.

The new model fits also identified IgG measurements were low before cases of very severe pneumonia at 6 mo and 18 mo. This difference was not identified in the analysis of post-disease IgG levels, suggesting that very severe pneumonia triggers an increase in anti-pneumococcal IgG levels (please see the response to Reviewer #2 point 2).

3. How do the protein-specific IgG levels correlate with serotype-specific IgG responses (capsule-specific) and do they show similar kinetics over time and with exposure? Is one a better predictor over the other?

We have now re-analysed data on the IgG response to five capsular polysaccharides, which were previously measured using an ELISA and analysed by Turner *et al* (2013 *Clin. Microbiol. Infect.* **19**(12):E551-558), as Supplementary Fig. 7. These data show that the kinetics of the response to

polysaccharides has some similarities to that of proteins: IgG levels fall from birth to six months, and then subsequently rise in some individuals (often only transiently), as a consequence of exposure to the corresponding capsule. However, many children exhibit no detectable response to three of the capsule polysaccharides (14, 19F and 23F) by 24 mo. This is consistent with a lack of exposure, or these molecules acting as T cell independent antigens, to which young children cannot mount a strong adaptive immune response. This provides additional evidence for the proposal that the weak response of 24 mo children to repetitive protein antigens, such as PclA, may represent such structures also acting as T cell independent antigens. However, there was no clear difference in the strength of the anti-capsular responses between the infant low and infant high immune clusters (Supplementary Fig. 7).

To test whether this divergence between the anti-protein and anti-capsular responses was a consequence of differences in immune activity, or an artefact of the different technologies used to assay IgG levels, the antibody binding levels to protein antigens assayed with ELISA by Turner *et al* were compared to similar proteins on the array. Despite the differences between the antigens, laboratories and technologies, a highly significant correlation in measured IgG binding was calculated across the majority of proteins. These data are now included in the manuscript as Supplementary Figure 1. The differences between the immune clusters, identified using the array data, could therefore be replicated using the ELISA measurements of IgG levels, as now shown in Supplementary Figures 5 & 6. This demonstrates the estimated differences between individuals' protein immunity levels are robust to the technology used to assay IgG levels, but these differences are not replicated when quantifying anti-capsular adaptive immunity. We have now edited the manuscript to state:

Across ages and ABT types, the IgG binding levels in the “infant high” cluster were 1.53-fold greater than those the more common “infant low” cluster (Supplementary Fig. 5). This ratio was estimated as a 1.49-fold difference using the ELISA antibody titres, confirming this distinction was not an artefact of the array analysis (Supplementary Fig. 5-6). By contrast, there was little evidence of a substantial difference in the anti-capsular IgG levels between clusters (Supplementary Fig. 7).

We also tested whether there was any evidence of anti-capsular antibodies protecting against pneumonia (Supplementary Fig. 63). No clear difference was observed between individuals who were diagnosed with pneumonia, and those who were healthy over the same age range, in agreement with Turner *et al*. However, this ELISA analysis studied a smaller cohort of children, and only analysed antibodies against five capsules. Hence it is difficult to draw broad conclusions about the relative importance of the overall response to all capsular polysaccharides, and that to the panproteome. We have therefore edited the text to state:

Furthermore, analysing previously-published data⁴⁵ on anti-capsular immunity from this cohort found no evidence such responses were protective against pneumonia (Supplementary Fig. 63).

4. The authors mention that the response is likely to be a reflection of total IgG but was this measured?

Unfortunately, the total IgG concentration of these samples was not measured. As a consequence of our reinterpretation of the function of ZmpB (please see response to point 6), we have now removed the relevant section of the manuscript.

5. If protein IgG responses do not block colonisation, then how would this fit into the larger picture of how to protect children against pneumococcal infections? As the authors rightly state, protein-based vaccines have had limited success.

This analysis demonstrates that infants generate a strong polyvalent adaptive immune response to pneumococcal proteins very early in life, yet this does not strongly inhibit colonisation. However, the new analyses we have added (Fig. 9) suggests that high maternal IgG may limit early colonisation, indicating that there may be an aspect of the mature adult IgG response that is effective at preventing acquisition of pneumococci. We now summarise this in the Results by stating:

By contrast, frequent colonisation in the first six months of life was associated with lower IgG levels in the umbilical cord sample, but higher IgG levels at 6 mo. Colonisation frequency ceased to be associated with differential IgG levels as the endogenous response strengthened with age (Fig. 9c).

In the Discussion, we suggest this may reflect the importance of broad maternal immunity against many DCL variants:

Consistent with this suggestion, previous work has suggested antibodies against a genotype's PspA and PspC variants are protective against carriage^{10,24,26}. This could explain how antibody-based protection against carriage might be genotype-specific^{13,16}: variation in DCL, even within strains^{51,82,83}, means that a newly-acquired pneumococcus may not be affected by IgA targeting previously-carried isolates. Correspondingly, the array demonstrated early infant IgG repertoires recognised few DCL protein variants compared to the broader immunity observed in umbilical cord and 24 mo samples. Hence both the protective effect of maternal immunity against frequent early colonisation, and the gradual reduction in carriage duration, density and frequency with age¹⁹, could be explained as a consequence of the accumulation of immunity against the range of circulating alleles of the DCL. Broad maternal immunity against DCL proteins could also explain the intergenerational correlations in IgG levels, as transplacentally-transmitted antibodies targeting a broad range of DCL proteins may accelerate the production of endogenous responses to other epitopes.

Furthermore, even if IgG are not protective against carriage in the nasopharynx, they may still be important at blocking progression to disease from carriage. This work demonstrates that children under 6 mo are capable of generating endogenous responses to pneumococcal proteins, and that lower IgG levels are associated with increased risk of pneumonia in the first months of life. Hence early vaccination with protein-based vaccines may be beneficial to children. Furthermore, the positive correlation of the levels of transplacentally-transmitted antibodies with those in 24 mo children suggests maternal vaccination may also be beneficial in protecting infants against pneumococcal disease. We have summarised these points in the Discussion

Hence this study not only demonstrates the ability of children <6 mo to generate endogenous antibodies to pneumococcal proteins, but also highlights the potential benefits of interventions that may stimulate these responses. Furthermore, IgG levels in the mother's blood were positively correlated with those of the infant at 24 mo, suggesting maternal immunisation could both protect neonates and boost responses throughout early childhood⁷⁶.

6. The authors state that ZmpB binds to the invariant region of IgG – would this suggest some Fc-mediated effector functions? Has/can this be measured? Some discussion of this might be worthwhile.

The reviewer raises an interesting point for discussion. We originally considered it likely that ZmpB bound immunoglobulins through an invariant part of their structure. This was inferred from the activity of the ZmpB orthologue ZmpA, and the strength of interactions between IgG molecules and the ZmpB N terminal probes, particularly in young children.

To test this proposition, we analysed the binding of human monoclonal IgG specific to capsular polysaccharides to the ZmpA and ZmpB probes on the array. These should not specifically recognise any pneumococcal proteins, but would be recognised by any non-specific antibody binding domains of ZmpB. However, the monoclonal IgG did not bind to any ZmpA or ZmpB probes non-specifically. Therefore our proposition that ZmpB is an antibody binding protein is incorrect. These data are now included in the manuscript as Fig. 3c.

Additionally, we plotted the frequency with which ZmpB N terminal probes were the proteins most strongly bound by IgG in each array dataset. If the IgG binding of ZmpB were independent of the antigen-binding sites, we would expect binding of ZmpB to be relatively high at the 6 mo timepoint, when the adaptive immune response to pneumococci is weakest, even if the absolute concentration of IgG is lower at this age. However, the relative binding to ZmpB was instead highest after the immune response matured between 12 and 24 mo. This is consistent with the recognition of ZmpB by IgG representing a genuine adaptive immune response to an epitope following exposure in carriage. These data are now shown in Fig. 3b. This poses the challenge of explaining how a strong response can emerge to the N terminus of all ZmpB variants by 6 mo, despite limited exposure to pneumococci by this age, and this part of the ZmpA orthologue being tethered closest to the cell surface.

We therefore undertook a re-analysis of existing data to formulate an improved explanation. The 279 N terminal amino acids of ZmpB are highly conserved across pneumococci, as demonstrated by alignment of the sequences of probes on the array (Supplementary Fig. 27). Using a single variant, an antibody fingerprinting study (Giefing et al 2008 J. Exp. Med. 205(1):117–131) found the strong immune reaction to ZmpB targeted the N terminal 495 amino acids. Hence it appears this conserved region is highly immunogenic, and therefore infants develop an adaptive immune response that can recognise all variants of ZmpB very early in life, which strengthens with age following repeated exposures.

The accessibility of this epitope appears to reflect an unusual aspect of ZmpB's post-translational processing. The LPXTG sortase attachment site lies within the conserved N terminal region of the protein, rather than the more typical C terminal location, meaning only the first ~75 amino acids are expected to be attached to the bacterial cell wall (Blue et al (2003) Infect. Immun. 71(9):4925-4935). The majority of the protein is instead predicted to be cleaved off and secreted, consistent with it being found at high concentrations in culture supernatants (Choi et al (2012) Diagnos. Microbiol. Infect. Dis. 72(4):318-327, Yamaguchi et al (2017) Virulence 8(8):1516-1524; Camilli et al 2006 Microbiol. 152(2):313-321). Both the small cell wall associated fragment, and the larger secreted fragment, appear to be immunogenic (Choi et al (2012) Diagnos. Microbiol. Infect. Dis. 72(4):318-327, Giefing et al 2008 J. Exp. Med. 205(1):117-131). Given the strong reduction in TNF- α levels associated with the loss of ZmpB expression (Blue et al (2003) Infect. Immun. 71(9):4925-4935), the larger ZmpB fragment therefore appears to function as a pro-inflammatory protein that is not physically associated with the pneumococcal cell, similar to pneumolysin.

Hence we have now redrafted the manuscript to include this novel interpretation of the data. The Results section states:

As 6 mo infants had developed a strong, broad response to all ZmpB variants, despite not having been exposed to many alleles, the IgG must recognise an epitope shared by all the representatives on the array. Despite the overall diversity of the ~2,000 amino acid (aa) protein, the N terminal 279 aa are highly conserved (Supplementary Fig. 27). Previous antibody fingerprinting analysis of a single variant found this was the region bound by immunoglobulins⁶⁰. Hence this highly immunostimulatory polypeptide is broadly recognised across pneumococci by the strongest emerging adaptive immune response of young children.

The Discussion section states:

This analysis suggests ZmpB also modulates the immune response. Unusually, the highly conserved and immunogenic N terminus contains an LPXTG sortase attachment site, meaning only the first ~75 aa are expected to be attached to the bacterial cell wall⁴³. The majority of the protein is instead predicted to be cleaved off and secreted, consistent with it being found at high concentrations in culture supernatants⁷⁸⁻⁸⁰. Hence the larger secreted ZmpB fragment appears to be a highly immunogenic, pro-inflammatory protein that is not physically associated with pneumococcal cells, consistent with the protein stimulating TNF- α signalling^{43,60,78}. The sequence conservation of the unstructured N terminal region⁵⁸ demonstrates its recognition by adaptive immunity is either inconsequential, or more probably beneficial, for ZmpB's function, as the highly variable sequence of the rest of protein suggests the likely catalytic domains of the metalloprotein are under selective pressure to evade immune recognition. This indicates ZmpB's unknown function is directed against mucosal antibody responses.

Reviewer #4:

The authors studied adaptive immune responses to pneumococcus in a prospectively-sampled cohort of 63 Maela infants during the first 2 years after birth, using integrated analyses of longitudinal panproteome array and pneumococcal genome data. With several dozens of Supplementary Figures, they extensively analyzed the valuable data and deeply assessed their results including detail parts. This study would well support the usefulness of the panproteome profiling in identifying heterogenous adaptive immune responses to pneumococcus variants. At the same time, their extensive presentations and descriptions likely make readers sidetracked from the main story. When considering Nature Communications is a general scientific journal, the current version of manuscript should be revised for its readership.

We thank the reviewer for their evaluation of the manuscript. In response to their comments, which echo those of Reviewer #2, we have sought to make the description of our analyses in the Results section more focussed by moving text to the Methods and Discussion sections, where appropriate.

Major points:

1. The authors concluded that “zinc metalloprotease B, exhibited characteristics of a high-affinity immunoglobulin-binding protein” (lines 18–19 of page2; and similar sentences in lines 7–8 of page 17). In my understanding from line 14 of page 5 to line 24 of page 6, this conclusion was derived mainly from Fig. S1B, which demonstrates that ZmpB N-terminal probes predominantly exhibited the maximum IgG binding among all probes. However, as the authors mention (lines 4–5 of page 6), N-terminus region of ZmpB is little exposed to the surface and thus unlikely functional as a physiological antigen within our body. Hence, there is a gap to conclude physiological function of ZmpB from their results. In addition, although they describe this conclusion even in Abstract, I do not think that this conclusion is an important finding in this study. Indeed, compared to the other diverse core loci (DCL) proteins, “the function of ZmpB remains unknown” (lines 19–20 of page 4). Nevertheless, the immunoglobulin-binding function could be easily assumed for ZmpB. I acknowledge that their demonstration of actual evidence is important in science, but this level of finding is not regarded as a significant advancement in modern Nature Communications. Therefore, they should remove or tone down this point.

We agree that the inference of ZmpB’s binding of the non-variable region of IgG required stronger supporting evidence. Therefore we tested whether ZmpB bound human monoclonal IgG that were known to be specific to capsular polysaccharides. This found no evidence of direct interactions. Therefore we have now concluded ZmpB does not non-specifically bind IgG, as detailed in the response to point 6 of Reviewer #3 above. Instead, the secretion of ZmpB proteins suggests IgG recognise highly-conserved N terminal epitopes of ZmpB, overturning our assumptions, and providing surprising new evidence of ZmpB’s function. We have now edited the manuscript accordingly.

2. The authors unexpectedly observed that the responses to ZmpB probes were less strongly correlated with age than non-antibody-binding target (non-ABT) probes (Fig. S1A), and interpreted that it was likely because of “saturating levels of antibody binding” (line 34 of page 5). I totally agree with their logical explanations. However, their interpretation also indicates that the signals of these

saturated probes (in their measurement condition) did not meet the sufficient data quality for assessment. Therefore, they should eliminate these probes during data processing and analyze the data without these probes throughout this study. Related to the above major point 1, it might be good to remove N-terminal probes for the other DCL proteins as well, depending on their choice of criteria. Of course, they must explicitly describe this kind of data exclusion in the Method section and Nature journal's Reporting Summary, but the data exclusion itself is reasonable. (Even after updating the results, it would be fair that they include the current Fig. S1–3 as supplementary information.)

Were our previous inferences regarding ZmpB function correct, we would have agreed with the reviewer, and adjusted our analysis as they suggested. However, as we now believe the recognition of the ZmpB N terminus represents a genuine response to exposure to a conserved antigen, we think it is important to capture this in our analysis, particularly given the strength of the response in young children.

3. The authors observed “substantial decline in IgG binding to all ABT types except ZmpB” (lines 31–32 of page 7) between birth and 12 months, followed by gradual rises of all non-ZmpB ABTs (Fig. 2A). What is a potential reason of this ZmpB exception? Again, if it was due to the N-terminal probes of ZmpB, they should have presented the result without these probes.

In our revised analysis, this exception appears to represent the highly immunogenic nature of the conserved ZmpB N terminus, which is readily accessible to antibodies, given its secretion into the extracellular milieu (beyond the capsule), and its unstructured nature. As this represents a genuine immune response, we have included these data from the N terminal probes in our analyses of the emerging antibody repertoire.

4. The authors should adjust all available potential confounding factors (as much as possible) within statistical tests/models throughout this study, especially because this study investigated only one cohort. For example, they addressed the possibility of several potential confounding factors for the relationships between IgG binding and ABTs per the factor (lines 8–27 of page 15; Figs. S52–56). However, it is better to present one single figure (Fig. 4) with these factors adjusted in statistical tests. Similarly, all linear mixed models they tried did not include the terms about these potential confounding factors, but they should have added these factors as fixed effects (without interactions). If the small sample size was the reason why they did not consider confounding factors within statistical tests/models, alternative method would be regressing out the potential confounding factors during data preprocessing and then using the residuals as signals throughout this study.

We agree with the reviewer that linear mixed effects models are the best way to quantify these effects on the IgG responses measured in the study. We can compare a series of simple mixed effect models that all feature age at time of sampling as a fixed effect and individual as a random effect, but differ in the inclusion of a fixed effect term representing an environmental factor (with no interaction terms, as specified by the reviewer):

Reference model	Comparison model	AIC _{ref} - AIC _{comp}	P value
IgG~Age+(1 Individual)	IgG~Age+Ethnicity+(1 Individual)	-5.6316318	0.2892983
IgG~Age+(1 Individual)	IgG~Age+Smoking+(1 Individual)	-0.7574895	0.1020863
IgG~Age+(1 Individual)	IgG~Age+Sex+(1 Individual)	-1.2699306	0.1439077
IgG~Age+(1 Individual)	IgG~Age+Birth Weight+(1 Individual)	-1.9985229	0.2943213
IgG~Age+(1 Individual)	IgG~Age+Fire+(1 Individual)	-1.3986455	0.1577775
IgG~Age+(1 Individual)	IgG~Age+Fuel+(1 Individual)	1.5835742	0.0258429

In all but one case, there is no evidence for the inclusion of the fixed effect improving the fit of the model. Only the model including house fuel type appears to improve the fit of the model, which would justify adjusting the IgG values. However, the reference level for the analysis of house fuel effect was a charcoal fire; no significant difference was found for children raised in homes with wood fires, with the only difference corresponding to households heated with a mixture of wood and charcoal (see model coefficient summary below). It seems unlikely that such a combination has a genuine biological effect bigger than either fuel type alone in their unmixed forms. Hence it is unlikely there would be any increase in accuracy by adjusting IgG measurements by environmental factors, in agreement with the univariate analyses shown in the original version of the study, which demonstrated limited association between IgG levels and environmental factors.

We address this in the manuscript using two approaches. First, we undertake a univariate association of the environmental and physiological factors with the assignment of individuals to the “Infant high” and “Infant low” immune clusters at 24 mo (Fig. 1b, 1c). This demonstrates there is no significant association between the immune clusters and these factors. We now state in the Results:

This clustering could reflect environmental or physiological heterogeneity within the cohort. However, the categorisation did not reflect differences in birth weight (Fig. 1b, Supplementary Fig. 8), nor was there a significant association with infant sex (Fig. 1c, Supplementary Fig. 9). Similarly, the four ethnicities represented in the cohort were associated with significant differences in IgG levels at some timepoints (Supplementary Fig. 10), but the most common identity, S'gaw Karen, was not congruent with the immune clustering. Furthermore, there was no evidence that the clustering was caused by air quality. Most homes were heated by wood or charcoal fires, but the type of fuel burned did not affect infant antibody responses (Supplementary Fig. 11). Additionally, 27.0% of the mothers smoked, and the children of these mothers had lower IgG responses to pneumococcal proteins at birth (Supplementary Fig. S12). However, the differences disappeared as the children aged, and maternal smoking was not associated with the immune clustering. Hence available epidemiological data could not explain the differences in the emergence of the antibody response.

Secondly, we used linear mixed effects models to test whether the heterogeneity in IgG levels between individuals observed in the data reflected any of the environmental or physiological factors. We repeated the analysis summarised in Fig. 2, but replaced the fixed effect of “immune cluster” with the environmental data categories listed above (Supplementary Table 3). Model comparisons using AIC values found none of the environmental factors explained the variation in IgG values as effectively as the immune clustering (Supplementary Figure 20). This is consistent with the heterogeneity between individuals not reflecting exogenous factors recorded in the epidemiological dataset, as originally concluded from the univariate analysis of IgG levels. Yet it improves the manuscript by providing a statistical basis for this conclusion that is consistent with the approaches applied in other analyses presented in the Results. We now summarise this in the manuscript by stating:

The model best representing the data included fixed effects that were consistent with antigen kinetics varying both by antigen type and the individual's cluster (Fig. 2c, Supplementary Fig. 18-19). Equivalent models that substituted the impact of immune cluster on IgG levels for environmental and physiological variables were less effective at explaining the differences in antibody responses, confirming the lack of association between these factors and the detected heterogeneity across individuals (Supplementary Fig. 20, Supplementary Table 3).

Minor points:

1. The authors frequently use “reproduced/reproducible across the cohort” (e.g., line 10 of page 2; line 1 of page 8). However, in life science, ‘reproduce a result’ usually indicates that the result is obtained from independent experiments/observations after the first experiment/observation. Hence, it would be better to use ‘consistently/commonly observed among the individuals’ instead, for example.

We have clarified our language in the manuscript as suggested by the reviewer.

2. “suggesting all four DCL modulate the host mucosal immune response” (lines 19–20 of page 2): the former part of this sentence says about only one DCL (ZmpB), and thus, until reading the Introduction

section, readers cannot understand why something about all four DCL proteins can be suggested from the former part.

We thank the reviewer for identifying the lack of clarity in this part of the Abstract, which has now been edited to state:

Quantifying such responses to the diverse core loci (DCL) proteins was complicated by cross-immunity between variants. In particular, the conserved N terminus of the DCL protein zinc metalloprotease B provoked the strongest early IgG responses. DCL proteins' ability to inhibit mucosal immunity likely explains continued pneumococcal carriage despite hosts' polyvalent antibody repertoire.

3. “Fig. S1” (line 23 and line 33 of page 5): the authors should exactly cite ‘Fig. S1A’ and ‘Fig. S1B’, respectively.

We have now made these references to figures more specific, as suggested.

4. “Fig. S1” (line 3 of page 6) would be a typo of ‘Fig. 1A’.

We thank the reviewer for highlighting this error, which has now been corrected.

5. “which shares structural similarities with PspC (Fig. S4)” (lines 16–17 of page 6): this way of figure citation is as if Fig. S4 shows structural (e.g., amino acid sequence-level) similarities between the ZmpB N-terminus and PspC. The authors should improve this sentence and citation position.

Figure S4 does show structural similarities between the N terminus of ZmpB and PspC: the PspC_subgroup_1 and PspC_subgroup_2 domains are ubiquitous across PspC proteins, and present in most ZmpB N terminal probe sequences. We have now clarified this in the revised manuscript by altering the text from “which shares structural similarities with PspC” to “and the ZmpB N terminus shares structural domains with the immunomodulatory PspC protein”.

6. “Hence the IgG binding for ... in the subsequent analyses.” (lines 19–24 of page 6): these sentences are about not logical consequences but just the criteria that the authors chose to use in this study. Therefore, these sentences are more suitable for the Method section. Basically, the Results subsection “ZmpB is a high-affinity antibody-binding protein” (line 14 of page 5 to line 24 of page 6) includes many supplementary information related to measurement/analysis conditions. Of course, all of them must be considered during the progress of study (and so, I acknowledge the authors' sincerity for science). However, the manuscript is not a notebook but a scientific article. Hence, the authors could simplify this subsection or move the details to Supplementary Note while citing the Method section.

We agree with the reviewer that this section was not an appropriate position in the manuscript to describe the technical details of the statistical analysis, which were unlikely to be of broad interest to the journal's readership. We have now extensively revised the start of the Results section, given our new

interpretation of the function of ZmpB (see above). In particular, the statistical analyses of variation in measurements between probes representing different parts of the same protein have been moved to the Methods section.

7. “Comparisons of the 377 IgG datasets identified extensive cross-sectional and longitudinal heterogeneity (Fig. S5, S6, S7)” (lines 27–28 of page 6): Figs. S5–7 are not good presentations to claim heterogeneity among the individuals. These figures are negative results in point of clustering individuals, but it is possible that it was just due to a bad projection from data space. (I believe the authors checked several parameters for the projection, though.) Basically, I do not think that Figs. S5–7 satisfy the roles of Supplementary Information for the findings of this study. The authors could remove these ‘negative’ results from the manuscript and deposit them to FigShare (lines 9–10 of page 24) as an original result.

While it is of course correct that these projections are inherently stochastic, we have allowed for a consistent number of iterations across the t-SNE analyses (detailed in the figure legends), to ensure they are similarly representative of the data. Analogously, we have used the default number of iterations recommended for the UMAP algorithm, to test whether it was able to identify structure that the t-SNE approach could not. Hence we believe the presented figures are representative of the data; we have made all the underlying raw data, and the code used to generate the figures, freely available to any reader who wishes to test this independently.

The guidelines of the journal state, “*We do not allow statements based on data that are not present in the manuscript or not published*”. Therefore, as these figures represent negative results that were used to inform the text of the manuscript, we think it is more appropriate to keep them in the supplementary materials than deposit them in FigShare.

8. “approximately half” (line 5 of page 7): it seems overestimated. Probably, were IgG binding values log₂-scaled? Then, the authors should add ‘log₂-scale’ to Y-axis title in all IgG binding figures.

All IgG measurements from the panproteome array are base 2 logarithmic measures of the change in fluorescence intensity. We have therefore changed all relevant graph axis labels to clarify this measurement as “(log₂ change in fluorescence)”. We have sought to clarify this in the Methods by adding the statement:

Therefore all IgG binding values from the array represent a change in the mean fluorescence intensity from the pre-serum background level, adjusted for batch effects, on a base two logarithmic scale.

The reviewer is correct that we slightly overestimated the difference between the clusters across all ABTs. Although it varies across antigen-binding protein types and ages, the overall ratio of IgG binding levels in the Infant high cluster, relative to the infant low cluster, is 1.53-fold. We have also now added an independent analysis that uses the previously-published measurements of IgG responses with ELISA, which estimates the ratio at 1.49-fold. The manuscript has now been edited to reflect this:

Across ages and ABT types, the IgG binding levels in the “infant high” cluster were 1.53-fold greater than those the more common “infant low” cluster (Supplementary Fig. 5). This ratio was estimated as a 1.49-fold difference using the ELISA antibody titres, confirming this distinction was not an artefact of the array analysis (Supplementary Fig. 5-6).

9. “Fig. S4” (line 6 of page 7) would be a typo of ‘Fig. S10’.

The reviewer is correct; we thank them for identifying this mistake, which has now been rectified.

10. “Consistent with individuals ... (Fig. 1C). Hence ... the child.” (lines 15–18 of page 7): these sentences have no logical ‘hence’ relationship. The pairs of umbilical cord and birth samples could be observed even if the unique property was derived from not the mother but the child. So, the authors should describe the latter sentence as a possible explanation.

We agree that the phrasing of this section of the manuscript was unclear. The birth blood sample was taken from the mother, and therefore the similarity with the umbilical cord sample must represent transplacental transmission from the mother to the child. We have therefore clarified this sampling throughout the Results section; in particular, we now state at the start of the Results section:

The panproteome array was used to assay blood samples from the births of 63 children (both from the mother and the umbilical cord), then from the child at 6, 12, 18 and 24 mo.

11. “Those belonging to the infant high cluster ... datasets (Fig. S11)” (lines 21–24 of page 7): the infant high and low are inverse between the sentence and Fig. S11. The authors should check the sentence or color coding of Fig. S11.

The reviewer is correct that Fig. S11 was a confusing representation of the data. We have now improved Fig. S11 (now Supplementary Fig. 16) to show the two clusters separately. Accordingly, we have edited the text to state:

Most sera from six-month-old children also co-clustered with later samples from the same individual (Supplementary Fig. 16). Hence the majority of children generate polyvalent responses to pneumococci by 6 mo, and the whole cohort had produced a detectably unique anti-pneumococcal IgG profile by 12 mo.

12. It is difficult to recognize the median line and violin shape in some violin plots (e.g., Fig. 1A). They should improve these plots, for example, by changing colors and placing the median line and violin to the top layer, like Fig. 4.

We thank the reviewer for this helpful suggestion; we have now standardised all the violin plots to show the violins as transparent (outlined in black) distributions, plotted over the appropriately-coloured points showing the raw data.

13. “Pearson correlation coefficient with umbilical cord sample” (Y-axis of Fig. 3B) would be a typo of ‘Pearson correlation coefficient with 6-months sample’.

We thank the reviewer for highlighting this error, which has now been corrected.

14. Figure 6 and Figs. S41–43: the authors could improve these figures, because half parts of these large figures are redundant, because information about row and column clustering is removed, and because the difference of values are not easily recognizable. Probably, they could summarize the correlation value distributions per variant with violin plot (with statistical tests for the distributions).

This comment refers to the plots showing the correlation between IgG responses to different variants of the four DCL (Fig. 8 and Supplementary Fig. 54-56 in the revised manuscript). The information on the clustering of proteins is fully represented by their ordering; the seriation method employed does not generate additional outputs (e.g. a dendrogram) that could be informatively added to the figure. Although the changes in correlations could be summarised by violin plots, this would lose substantial information on both the individual pairwise similarities, and the overall structure of cross-immunity. As some of the individual relationships are important for understanding the relationship between exposure and adaptive immune response for DCL variants, we think it is better to use this correlogram, which is a common approach for representing a correlation matrix.

15. “shows” (line 4 of page 43) → ‘show’

We have now made this correction.

16. Fig. S1B: because many points overlap with each other, it would be better to describe how many ZmpB N-terminal probes actually exhibited the maximum signals among all probes.

The reviewer is correct that this figure was unclear. We have therefore increased the transparency of the points in Fig. S1B (now Supplementary Figure 24). We have also calculated that the polypeptide binding IgG most strongly was a ZmpB N terminal probe in 40.3% of samples, and IgG binding to such probes was within 10% of the maximal observed IgG binding in 91.5% of samples. We have added a plot (Fig. 3b) showing the distribution of proteins to which IgG binding was maximal across all datasets, separated by age at time of sampling. This demonstrated how rapidly the responses to the ZmpB N terminus developed. We have now edited the manuscript to state:

The IgG responses to ZmpB were atypical in changing little with age, due to consistently high binding of all variants’ N terminal regions (Fig. 3a, Supplementary Fig. 23). ZmpB N terminal probes provoked the

strongest IgG interaction in 40% of serum samples (Supplementary Fig. 24), including the majority of datasets at 12 mo (Fig. 3b).

This analysis was helpful in reinterpreting the interaction between ZmpB and the immune system, and so we thank the reviewer for suggesting this improvement.

REVIEWER COMMENTS

Reviewer #2 (Remarks to the Author):

The authors have thoroughly addressed my comments in the revision.

Reviewer #3 (Remarks to the Author):

Thankyou to the authors for their detailed revisions, which have improved the manuscript.

Reviewer #4 (Remarks to the Author):

The authors addressed most of my concerns (although they chose to ignore some minor points). However, they did not fully understand my concern of the major point #4, which must be further addressed before publication.

Major point:

1. As a response to my previous major point #4, the authors denied the effects of potential confounding factors (i.e., birth weight, infant sex, mother's ethnicity, type of fuel in households, and mother's smoking) on their findings (i.e., the associations between IgG binding levels and sampled ages), based on no statistical significances in the IgG differences between each factor's categorical levels. First of all, however, no statistical significance does not indicate that the null hypothesis is true. Additionally, this logic ignores multivariable relationships within the data. Therefore, what I originally requested was testing the authors' findings while adjusting the potential confounding factors.

More concretely speaking, the authors should test the associations between IgG binding levels and sampled ages by beta1 coefficients in a generalized linear model (GLM): $IgGbindingLevel \sim \beta_1 \cdot SampledAge + \beta_2 \cdot BirthWeight + \beta_3 \cdot InfantSex + \beta_4 \cdot MotherEthnicity + \beta_5 \cdot FuelType + \beta_6 \cdot MotherSmoking$. Here, we do not mind whether beta2–6 coefficients are 0 or not. This model may overestimate the effects of potential confounding factors, but, if beta1 coefficients are not 0 even this overestimated model, it provides more robust conclusions (i.e., conservative approach). Of note, the linear mixed model (LMM) by adding random effects to the above GLM would be alternative choice, although what Supplementary Figs. 8–12 tested seem corresponding to the above GLM.

Similar things are applicable to their LMM analyses. After the authors decided a parsimonious model based on AIC, they should add the potential confounding effects as fixed effects and test whether the

beta-coefficients of target variables are 0 or not. Again, this added model may overfit data, but would conservatively support their claims.

Note that I understand some other studies utilize the authors' current logic, but it's weaker and usually used as a compromise when the sample size is not sufficient in point of statistical power. In such case, I would recommend regressing out the potential confounding factors during data preprocessing.

NCOMMS-23-35488B: Response to Reviewers

All reviewer comments are reproduced in **boldface**. Our responses are in regular typeface. The code for the additional analyses we have conducted as part of this response is available from the associated Github repository.

Reviewer #1:

No comments received.

Reviewer #2:

The authors have thoroughly addressed my comments in the revision.

We are grateful for the reviewer's help in evaluating our manuscript.

Reviewer #3:

Thankyou to the authors for their detailed revisions, which have improved the manuscript.

We thank the reviewer for improving the manuscript with their suggestions.

Reviewer #4:

The authors addressed most of my concerns (although they chose to ignore some minor points). However, they did not fully understand my concern of the major point #4, which must be further addressed before publication.

We thank the reviewer for their thorough and prompt evaluation of our manuscript. We are sorry the reviewer felt we ignored some of their points. We sought to address each of their minor points, making modifications where we felt it appropriate, and explaining our reasoning in cases where we disagreed. Hopefully we can fully address any remaining concerns in this response.

Major point:

1. As a response to my previous major point #4, the authors denied the effects of potential confounding factors (i.e., birth weight, infant sex, mother's ethnicity, type of fuel in households, and mother's smoking) on their findings (i.e., the associations between IgG binding levels and sampled ages), based on no statistical significances in the IgG differences between each factor's categorical levels. First of all, however, no statistical significance does not indicate that the null hypothesis is true. Additionally, this logic ignores multivariable relationships within the data. Therefore, what I originally requested was testing the authors' findings while adjusting the potential confounding factors.

The reviewer is of course correct that not being able to reject the null hypothesis does not mean these factors have no effect. However, we did not deny the effects of confounding factors, but rather tested a series of models that demonstrated they did not substantially

affect the results. Our observations are consistent with the factors affecting antibody responses under scenarios such as:

- Such effects may be so small that they are not measurable, relative to the biological and technical noise of the study, resulting in the data lacking the necessary precision to identify the correct scale of the effect.
- Such effects may be smaller than the effects of unmeasured variables (e.g. host genetic variation), meaning that adjustments cannot be made accurately used a fixed effects approach.
- The interactions between such effects (e.g. as described by the underlying causal network) may mean their impact upon antibody levels cannot be suitably described by fixed effect terms that are constant across the cohort

In all our models, we correct for inter-individual variation using a random effect term. This both corrects for both the measured, and unmeasured, characteristics of each individual. For example, the random effects terms could, in principle, not just correct for the effect of maternal smoking in a subset of the cohort, but also quantitatively adjust for differing levels of smoking between mothers (on which we do not have information). In addition, the random effects terms account for different combinations of multivariable relationships, which may be unique to an individual within the cohort. The disadvantage of this approach is that we cannot quantify the impact of the individual effects on antibody levels. However, this is not a problem for our analysis, which is focussed on other factors. Therefore we believe the modelling results will be robust to variation across the cohort, even in the absence of fixed effects terms estimating the effect of individual measured properties.

More concretely speaking, the authors should test the associations between IgG binding levels and sampled ages by beta1 coefficients in a generalized linear model (GLM):
 $IgG_{bindingLevel} \sim \beta_1 \cdot SampledAge + \beta_2 \cdot BirthWeight + \beta_3 \cdot InfantSex + \beta_4 \cdot MotherEthnicity + \beta_5 \cdot FuelType + \beta_6 \cdot MotherSmoking$. Here, we do not mind whether beta2–6 coefficients are 0 or not. This model may overestimate the effects of potential confounding factors, but, if beta1 coefficients are not 0 even this overestimated model, it provides more robust conclusions (i.e., conservative approach). Of note, the linear mixed model (LMM) by adding random effects to the above GLM would be alternative choice, although what Supplementary Figs. 8–12 tested seem corresponding to the above GLM.

We fitted the linear model proposed by the reviewer, with an extra term for whether or not there was a fire in the household, resulting in six fixed effects that potentially confounded our analysis of changes with age (maternal ethnic group; maternal smoking behaviour; infant sex; infant birth weight; house fire; house fuel). Multiple beta estimates were significantly different from zero in this analysis (Fig. 1).

Figure 1 Inference of beta parameter values using a linear model containing only fixed effects to estimate the impact of age and six other potentially-confounding factors on immunoglobulin G levels. Each point represents the maximum likelihood estimate, and the error bars represent the 95% confidence intervals. Points are coloured red if the estimate is below zero, and blue if the estimate is above zero.

When compared to an equivalent model that only includes age as a fixed effect, the more complex model is favoured by the AIC. Therefore, in a linear or generalised linear modelling framework, it would be justifiable and important to adjust for the variation between individuals as extensively as possible, as the reviewer suggests.

However, if we add the random effects term for each individual, only one of the beta estimates for the potentially confounding factors significantly differs from zero (based on the 95% confidence intervals; Fig. 2). By contrast, the trend of interest (changes relative to age) is robust to the altered modelling approach.

Figure 2 Inference of beta parameters using a mixed effects model containing the fixed effects from the analysis shown in Fig. 1, and a random effect per member of the cohort. Data are shown as described in Fig. 1.

When this mixed effects model was compared with the equivalent model that only includes age as a fixed effect, and individual as a random effect, the AIC favours the simpler model. Therefore, in a mixed effects modelling framework, it is not necessary to include fixed effects to account for the variation across the cohort. This is due to the adjustment factor of the random effect, estimated for each individual. This value is crucial to all the modelling analyses, because it accounts for the non-independence of measurements from a single individual.

In the mixed effects analysis (Fig. 2), the only non-age fixed effect that was found to significantly deviate from zero was the positive effect of using a mixture of fuel types in a house fire on antibody responses (consistent with the analysis presented in the original response to the reviewer). As charcoal fuel was the reference level, set to zero, and wood was estimated to negatively affect antibody levels, it seems implausible that a mixture of wood and charcoal would be genuinely beneficial (note that only four households in the cohort used a mixed fuel fire). Instead, this is likely to be a false positive resulting from an effect independent of fuel type that affects this small subsample of families. Such an effect may be due to an unmeasured (e.g. cultural or genomic) aspect of the cohort. This highlights a risk of adjusting the data using the approach described by the reviewer: we may incorrectly change values due to the limitations of the data available on the members of the cohort.

Every model we fitted (detailed in Supplementary Tables 2, 3, 7, 9 and 10) included a random effects term adjusting for variation between individuals. Therefore, given the consistent use of a mixed effects modelling framework, the per-individual adjustment is

sufficient to account for variation across the cohort, when employing standard statistical approaches to model selection. We address the point regarding the robustness of inferences below.

The supplementary figures to which the reviewer refers show univariate analyses of the raw data. These are inherently confounded by the structure of the cohort, including the variation between individuals shown in Fig. 1a. Nevertheless, the value of these graphs is to show the distribution of the unprocessed IgG response values, such that the reader is able to easily gain an intuition for the differences associated with variables of interest. Partly, this allows the reader to understand the magnitude of different effects on responses (e.g. comparing the differences between immune clusters, differences between children of different sexes, or between exposed and unexposed individuals), which is not possible if data have already been adjusted for multiple factors.

No major conclusions were inferred based on these plots of raw data alone. Where these plots did suggest differences between groups (e.g. between exposed and unexposed individuals, or following very severe pneumonia infections), the difference was quantified using a mixed effects modelling framework with a random effect to adjust for differences between individuals. Nevertheless, these plots of the raw data were informative in the formulation of appropriate model structures, and complemented the less intuitive, but more formal, multivariate analysis of antibody levels. Therefore they are present in the supplementary materials in accordance with the journal's requirement that all data on which the study is based should be included in the manuscript.

In cases where these plots did not suggest significant differences between groups (e.g. between sexes), this was validated using independent approaches where possible (e.g. Fig. 1c). Yet if these values were to be adjusted, as suggested above, it would not be possible to display the raw data for the differences included in the adjustment (i.e. adjusting for a child's sex would not allow interrogation of the underlying differences in the raw data between male and female children). Furthermore, if previously-undetected differences were to emerge following the adjustment, these inferences would not be robust. This is because the wide confidence intervals of the adjustment values involved span zero (i.e. no effect) in almost every case in the mixed effects analysis above, or, in the case of the effect of fuel type, are highly unlikely to be correctly attributing variation in antibody levels to the true cause. Hence we think it is valuable to retain the graphs showing the raw data for the reader, as a complement to the more complex statistical analyses.

Similar things are applicable to their LMM analyses. After the authors decided a parsimonious model based on AIC, they should add the potential confounding effects as fixed effects and test whether the beta-coefficients of target variables are 0 or not. Again, this added model may overfit data, but would conservatively support their claims.

Criteria such as AIC are used to identify the factors that significantly improve the fit of a model. If we use AIC as a criterion to select a model structure, it is inconsistent to add extra fixed effects terms that themselves would be excluded from a model structure using the same approach, based on the analysis presented in Fig. 2. Therefore we think it is

appropriate to retain the current, common approach of selecting an appropriate model using the AIC.

We appreciate it would be informative to test whether the reviewer’s suggested change to our approach affected our inferences from the data. Therefore we quantified the effect of six fixed effect terms (maternal ethnic group; maternal smoking behaviour; infant sex; infant birth weight; house fire; house fuel) on the analysis of the association of IgG levels with pneumonia episodes since the previous serum sample (Supplementary Table 9). We fitted model 2 from Supplementary Table 9 to data on antigens to which exposure was universal across the cohort at 12 months old (mo). We fitted model 5 from Supplementary Table 9 to data on antigens to which exposure was partial across the cohort at 12 mo. Both models were then refitted with the six extra fixed effects terms.

Figure 3 shows the main results, the differences in IgG binding after a pneumonia episode at different ages (not corrected for the baseline, as they are in Fig. 9). The top row shows the fit of model 5 to data from antigens to which only a subset of the cohort was exposed by 12 mo; the bottom row shows the fit of model 2 to data from antigens to which all members of the cohort were exposed by 12 mo. The left column shows the values estimated with the six potentially confounding effects added to the model; the right column shows the results from the original model formulations, lacking these fixed effect terms. In each case, the estimates were near-identical, whether or not the six potentially confounding fixed effects were included.

Figure 3 Barplots showing the relative change in IgG binding with age in individuals suffering from an episode of pneumonia in the previous six months, estimated with a mixed effects model. The plots on the top row show the data for antigens to which only a subset of individuals were exposed by 12 mo; the bottom row plots show the same estimates for antigens to which all individuals were exposed by 12 mo. The right column shows the original analysis

included in the manuscript. The left column shows the results for models including the six extra fixed effect terms.

These models also estimated the relationship between antibody levels in the maternal blood and those in the infant for each antibody-binding target (Supplementary Table 9). The distributions of these values were essentially identical (Fig. 4), regardless of whether the six potentially-confounding fixed effect terms were included. This again demonstrates that our conclusions would not change if we were to make the alterations suggested by the reviewer.

Figure 4 Density plots showing the strength of the association between maternal and infant antibody levels across antigens, estimated using mixed effects models. The plots on the top row show the data for antigens to which only a subset of individuals were exposed by 12 mo (model 5); the bottom row plots show the same estimates for antigens to which all individuals were exposed by 12 mo (model 2). The right column shows the original analysis included in the manuscript. The left column shows the results for models including the six extra fixed effect terms.

Additionally, the models without the six added fixed effects were favoured by AIC for both sets of antigens. Therefore the addition of the six extra fixed effects is both difficult to justify theoretically, and had little effect on the outputs of the models.

Furthermore, introducing extra terms into models does not necessarily make them more conservative, or their inferences more robust. The effect depends on the structure of the underlying causal network that describes the interactions between the factors (Pearl, 2014, Am. Stat. 68(1):8-13). In the cohort we have studied, the environmental and physiological factors are linked together in such a causal network. For instance, there are links between smoking behaviour and maternal ethnicity, and also birth weight. In such circumstances,

adding extra terms (particularly ones that do not actually detectably affect antibody levels) risks the false positive inference of an association between variables. Without knowing the full set of causal interactions between the terms, using fixed effects to account for the impact of different variables is not necessarily a conservative approach.

Nevertheless, the random effects term for each individual essentially corrects for the interdependence of these factors, by assigning each person a unique adjustment value that is particular to their genetics, physiology and environment. Hence the model structure we use in the analysis corrects for variation across the cohort, but minimises the risk of false positive inferences of environmental or physiological differences affecting antibody responses. Therefore we believe it is the most conservative approach to the statistical modelling of the data.

Note that I understand some other studies utilize the authors' current logic, but it's weaker and usually used as a compromise when the sample size is not sufficient in point of statistical power. In such case, I would recommend regressing out the potential confounding factors during data preprocessing.

We believe it is common best practice to take measures to avoid overfitting with complex models, as recommended by the Points of Significance series in *Nature Methods* (Lever et al 2016 Nat. Meth. 13:703-704). As this is the first such dataset of its type, there is no equivalent validation dataset, as would be ideal for the test-train-validate workflow recommended by Lever *et al.* Therefore we rely on long-established, commonly-used criteria, such as AIC, which were designed to avoid overfitting of unnecessarily complex models. We do not consider it a sign of weakness in the analysis, as AIC-based approaches are implemented in many common statistical packages (e.g. stepAIC in the R package MASS; information criteria methods in the python package sklearn) and broadly used across many research disciplines.

With a dataset of this complexity, it is not possible to completely regress out the effects of variation across the cohort. There are many variables, measured and unmeasured, that likely affect immune development. Therefore we feel the best approach is to present complementary analyses in which the raw data are shown, and then subject to multivariate analyses that correct for each individual's unique characteristics using a random effect term. However, as the random effects terms are independently fitted for each model, determined by both the fixed effects terms and population-wide distribution of antibody responses, it is not possible to use these to provide a single adjusted antibody binding level that can be plotted across all proteins. We hope our manuscript contains an appropriate combination of suitably-adjusted statistical comparisons, and transparent presentation of the underlying unprocessed data.

REVIEWERS' COMMENTS

Reviewer #4 (Remarks to the Author):

The authors completely resolved my concerns, including ones derived from my misunderstanding about their statistical design. I appreciate their attentive explanations.